# SpikeLLM: Scaling up Spiking Neural Network to Large Language Models via Saliency-based Spiking

**Xingrun Xing[1,2,3], Boyan Gao[4], Zheng Liu[3]\*, David A. Clifton[4], Shitao Xiao[3],**
**Wanpeng Zhang[5], Li Du[3], Zheng Zhang[3], Guoqi Li[1,2]\*, Jiajun Zhang[1,2]\***

[1] School of Artificial Intelligence, University of Chinese Academy of Sciences
[2] Institute of Automation, Chinese Academy of Sciences
[3] Beijing Academy of Artificial Intelligence   [4] University of Oxford   [5] Peking University
`zhengliu1026@gmail.com, guoqi.li@ia.ac.cn, jjzhang@nlpr.ia.ac.cn`

## Abstract

Recent advancements in large language models (LLMs) with billions of parameters have improved performance in various applications, but their inference processes demand significant energy and computational resources. In contrast, the human brain, with approximately 86 billion neurons, is much more energy-efficient than LLMs with similar parameters. Inspired by this, we redesign $7\sim70$ billion parameter LLMs using bio-plausible spiking mechanisms, emulating the efficient behavior of the human brain. We propose the first spiking large language model, SpikeLLM. Coupled with the proposed model, two essential approaches are proposed to improve spike training efficiency: Generalized Integrate-and-Fire (GIF) neurons to compress spike length from $T$ to $\frac{T}{L}\log_2 L$ bits, and an Optimal Brain Spiking framework to divide outlier channels and allocate different $T$ for GIF neurons, which further compresses spike length to approximate $log_2 T$ bits. The necessity of spike-driven LLM is proved by comparison with quantized LLMs with similar operations. In the OmniQuant pipeline, SpikeLLM reduces 11.01% WikiText2 perplexity and improves 2.55% accuracy of common scene reasoning on a LLAMA-7B W4A4 model. In the GPTQ pipeline, SpikeLLM achieves direct additive in linear layers, significantly exceeding PB-LLMs. Our code is publicly available at `https://github.com/Xingrun-Xing2/SpikeLLM`.

## 1 Introduction

Recent Artificial Neural Networks (ANNs) have shown scaling up Large Language Models (LLMs) (Brown et al., 2020; Touvron et al., 2023b; Zhang et al., 2022a; Le Scao et al., 2023) can be one of the most potential techniques to access Artificial General Intelligence. However, despite these unprecedented and promising achievements, steering LLMs imposes a tremendous burden in terms of energy costs and computational requirements. For instance, running inference on the LLAMA-2-70B model requires $2\times$ A100-80 GPUs, and each energy consumption is 400W. This creates a significant obstacle for deploying LLMs to real-world applications, especially where limited battery capacity and memory size are critical, such as in mobile devices (Xing et al., 2025). To lower these barriers and broaden the applications of LLMs, we focus on energy-efficient artificial intelligence.

Compared with ANN-based LLMs, human brain nervous systems achieve superior intelligence with much less energy consumption (Gerstner et al., 2014; Izhikevich, 2003) and a comparable number of neurons, approximately 86 billion. For several decades, the brain-inspired computing (BIC) field (Mehonic & Kenyon, 2022; Zhang et al., 2020) focuses on mimicking the biological nature of the human brain to develop more efficient and general AI algorithms (Maass, 1997) and physical platforms (Schuman et al., 2022; Roy et al., 2019; Pei et al., 2019). Among these, spiking neural networks (SNNs) (Maass, 1997; Gerstner et al., 2014) are particularly notable for their biological

---

\* Corresponding author

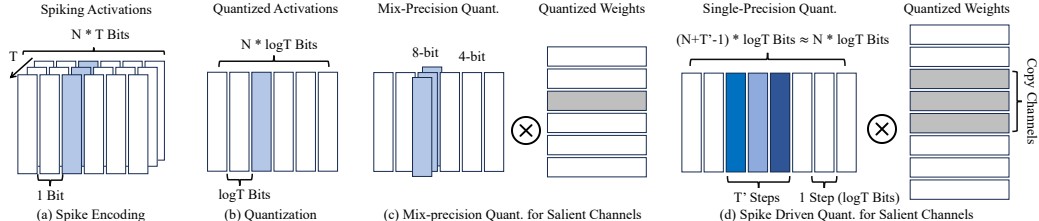

Figure 1: Different encoding methods. In (a, b), the activation has N channels; each value has T quantization levels. Given salient channels (in blue), mix-precision methods (c) are deployment unfriendly. In spike-driven methods (d), we expand salient channels by spiking dynamics to realize single precision quantization, where T' is spiking steps in salient channels.

Table 1: Comparason of encoding efficiency. L indicates quantization levels in each step, where $L \in [1, T]$. The accuracy and perplexity are evaluated in the common scene reasoning and WikiText with the LLAMA-7B, W4A4 setting (Appendix A.2).

| method | Steps | Bits/Step | Bits | Quant-Levels | Acc. ↑ | PPL ↓ |
|---|---|---|---|---|---|---|
| IF-SNN | $T$ | 1 | $T$ | $T$ | – | – |
| GIF-SNN | $\frac{T}{L}$ | $log_2 L$ | $\frac{T}{L} log_2 L$ | $T$ | – | – |
| Quant-ANN | 1 | $log_2 T$ | $log_2 T$ | $T$ | 49.93 | 11.85 |
| SpikeLLM | $\approx 1$ | $log_2 T$ | $\approx log_2 T$ | $\approx T$ | 52.48 | 9.98 |

plausibility and binary event-driven efficiency (Yin et al., 2021; Schuman et al., 2022). Despite their potential efficiency advantage, recent SNNs face two significant bottlenecks: (i) Firing Rate Encoding (Bu et al., 2023; Li et al., 2021; Deng & Gu, 2021) in existing SNNs is inefficient in capturing adequate semantic information. As shown in Table 1 and Fig. 1 (a, b), to express $T$ quantization levels, a typical Integrate-and-Fire (IF) (Liu & Wang, 2001; Barbi et al., 2003) neuron requires $T$ time steps to encode full information, while quantization only requires $log_2 T$ bit digits in one step. (ii) Inefficient Optimization. Direct training (Neftci et al., 2019; Wu et al., 2018) SNNs need gradient estimation in BackPropagation Through Time (BPTT) because of non-differentiable spiking dynamic; the ANN-SNN conversion (Han et al., 2020; Hao et al., 2023; Bu et al., 2023; Li et al., 2021) often requires much more inference steps to simulate ANNs, both of which are impractical for scaling up to LLMs. These challenges have kept SNNs relatively small (under 1 B parameters) and hinder their ability to scale to the complexity of the human brain.

This work aims to scale up SNNs to large language models or even part of the human brain nervous system with 86B neurons. Towards this goal, we propose both a micro spiking neuron design to improve spike efficiency and a macro saliency-based spiking design to drive spiking neurons. Currently, the primary approach to encode real-valued features to limited binary digits is model quantization. Given $T$ quantization levels, quantization functions can efficiently encode features into $log_2 T$ binary digits in one step but encounter significant quantization errors for outliers or salient values in low-bit conditions (Xiao et al., 2023; Lin et al., 2023b). Inspired by both quantized ANNs and SNNs, we introduce a hybrid encoding method of quantization and spike firing rate encoding, termed Generalized Integrate-and-Fire (GIF) neurons. In each spiking step, we merge $L$ steps in IF neurons and encode as $log_2 L$ bits as quantization; across different steps, we keep the recursive spiking dynamics. Compared with IF neurons, GIF compresses spike length from $T$ bits to $\frac{T}{L} log_2 L$ bits, where $L$ is the quantization levels in each step. Compared with quantization, GIF neurons maintain recursive encoding advantages to accurately quantize salient channels with multisteps, which gives the potential to exceed quantization. To further determine channel-wise spiking steps in GIF neurons, as shown in Fig. 1 (d), a saliency-based spiking mechanism is proposed to divide and conquer salient channels and almost non-salient ones. And then, we allocates multistep spiking to encode salient channels and one-step spiking for others, which further compresses to approximate $log_2 T$ bits over all channels. Unlike mix-precision quantization in Fig. 1 (c), saliency-based spiking expands salient channels with GIF neurons and equals to single-precision quantization in Fig. 1 (d).

To detect salient channels, we propose a framework named Optimal Brain Spiking (OBSpiking), which is a weight-activation generalization of the classic Optimal Brain Surgeon (OBS) (Hassibi & Stork, 1992) for model pruning. The key concept of our method is distinguished approximations of weight and activation Taylor expansion to calculate their saliency. In detail, the first-order gradient

and second-order Hessian metrics are leveraged for activations and weights respectively. For every matrix multiplication in LLMs, GIF neurons can be viewed as generalized quantizers to quantize salient and other channels based on the saliency rank generated by the OBSpiking framework.

To evaluate the necessity of the Spiking LLMs, we integrate the proposed spiking mechanism with the most classic LLM quantization pipelines including the Omniquant (Shao et al., 2023) and GPTQ, named SpikeLLM. For weight-activation quantization (Shao et al., 2023), we observe significant performance improvements in generation and common scene reasoning. For weight-only quantization, GIF neurons further quantize ternary weights with the GPTQ (Frantar et al., 2022) pipeline, achieving direct addition networks. SpikeLLM exceeds PB-LLM (Shang et al., 2023) dramatically.

Our contributions are summarised as follows:

- We first scale up spiking neuronal dynamics to 7∼70 billion parameters, promoting SNNs to the era of LLMs. The necessity of introducing a spiking mechanism to LLMs is proved by the comparison with quantization methods.

- We propose a Generalized Integrate-and-Fire neuron and Optimal Brain Spiking framework as a general alternative to traditional quantizers. The efficiency issue of firing rate encoding in SNNs is addressed by spike length compression.

- The proposed GIF neuron and OBSpiking can be implemented by both general matrix multiplication (GEMM) and binary event-driven operations, providing the first practical design of SNNs with more than tens of billion parameters.

## 2 RELATED WORKS

**Brain-Inspired Computing.** The Brain-Inspired Computing (BIC) (Mehonic & Kenyon, 2022; Zhang et al., 2020) field focuses on building up bio-plausible and general fundamental theories, algorithms, software (Fang et al., 2023), and hardware platforms (Schuman et al., 2022; Roy et al., 2019; Pei et al., 2019) inspired by the structures and functions of biological nervous systems like the human brain. Spiking neural network (SNN) (Roy et al., 2019; Maass, 1997) is one of the most popular BIC algorithms which embeds biological spiking neural dynamics in each single neuron (Yin et al., 2021; Schuman et al., 2022). Promoted by the development of both deep learning and advanced biological neuron science, recent SNNs focus on the deep residual learning (Fang et al., 2021), self-attention (Yao et al., 2023; Zhou et al., 2023), normlization (Zheng et al., 2021), as well as biological learning rules (Payeur et al., 2021), structures (Pham et al., 2021) and energy efficiency (Schuman et al., 2022). In optimization, recent SNNs apply ANN-SNN conversion (Han et al., 2020; Hao et al., 2023; Bu et al., 2023; Li et al., 2021) or direct training (Neftci et al., 2019; Wu et al., 2018) techniques. Most SNNs focus on the computation vision field; language-oriented SNNs are almost less than 1 billion parameters, for example, SpikeLM (Xing et al., 2024b), SpikingBERT (Bal & Sengupta, 2024; Lv et al., 2023; Su et al., 2024), and SpikeGPT (Zhu et al., 2023). How to scale up bio-inspired spiking mechanisms to billions of parameters has become a valuable issue.

**Model Quantization.** Model quantization aims at reducing the bit-width of weights or activations to accelerate network inference exemplified by the recent quantization works including Post-Training Quantization (PTQ) (Xiao et al., 2023; Frantar et al., 2022), Quantization Aware Training (QAT) (Liu et al., 2023b), and calibration training methods (Shao et al., 2023). For small neural networks, QAT methods (Esser et al., 2019; Liu et al., 2022; Xing et al., 2024a; 2022b;a), benefit from the specifically designed training protocol, achieving outstanding performance. However, these methods are not practical in the LLM setting. In contrast, PTQ methods including GPTQ (Frantar et al., 2022), GPTQ-ada (Heo et al., 2023), SpQR (Dettmers et al., 2023), OWQ (Lee et al., 2023), AWQ (Lin et al., 2023a), and PB-LLM (Shang et al., 2023) tailored for LLM are weight-only quantization; SmoothQuant (Xiao et al., 2023) and RPTQ (Yuan et al., 2023) achieve weight-activation quantization. Besides, LLM-QAT (Liu et al., 2023b), QA-LORA (Xu et al., 2023), and calibration based methods including Omniquant (Shao et al., 2023), QLLM (Liu et al., 2023a), AffineQuant (Ma et al., 2024), and QuaRot (Ashkboos et al., 2024) achieve higher performance. However, current quantized LLMs encounter significant quantization errors in outlier or salient channels as discussed in Section 3.2. Althrough more recent rotation quantization (Ashkboos et al., 2024) achieves outlier-free, there needs additional matrix multiplications that can not be absorbed.

## 3 PROBLEM FORMULATION

In this section, we first introduce bio-inspired Spiking Neural Networks (SNNs) and explore the potential of replacing traditional quantized Large Language Models (LLMs) with spiking LLMs. Additionally, we address the challenges of outlier and salient value quantization, which motivate the introduction of the spiking mechanism.

### 3.1 SPIKING NEURONAL DYNAMICS

SNNs can be considered ANNs by adding biological spiking neuronal dynamics in each neuron. Without loss of generality, the biological soma dynamics are approximately modeled by the first- or higher-order differential equations. The IF neuron is a first-order approximation of soma dynamics, combining the advantages of bio-plausibility and efficiency, which can be represented by:

$$\mathbf{v}(t) = \mathbf{v}(t-1) + [\mathbf{x}^{(\ell-1)}(t) - \min(\mathbf{x}^{(\ell-1)})] - \mathbf{s}^{(\ell)}(t-1)V_{th}, \tag{1}$$

$$\mathbf{s}^{(\ell)}(t) = \begin{cases} 0, & \text{if } \mathbf{v}(t) < V_{th} \\ 1, & \text{if } \mathbf{v}(t) \geq V_{th} \end{cases}, \tag{2}$$

$$\mathbf{x}^{(\ell-1)}(t) = \mathbf{s}^{(\ell)}(t)V_{th} + \min(\mathbf{x}^{(\ell-1)}), \tag{3}$$

where the IF neuron encodes a binary spike in each step $t$ for a duration of $T$. In Eq.1, the membrane potential $\mathbf{v}(t)$ accumulates current input $\mathbf{x}^{(\ell-1)}(t)$ to the last time step $\mathbf{v}(t-1)$ to simulate the charging process in soma. A subjection reset is applied to subtract the spiking values from the membrane potential $\mathbf{v}(t)$. In Eq.2, if $\mathbf{v}(t)$ exceeds a certain firing threshold $V_{th}$, the neuron is fired and encodes the spike $\mathbf{s}^{(\ell)}(t)$ as 1; otherwise, encodes as 0. Thus, previous SNNs take the IF neuron as an activation quantizer, which encodes full-precision activation as 1-bit output per spiking step in Eq. 3. Different from SNNs with ReLU activations, to encode the negative values in transformers, we encode $\mathbf{x}^{(\ell-1)}(t) - \min(\mathbf{x}^{(\ell-1)})$ by spike firing rate, where $\min(\mathbf{x}^{(\ell-1)})$ is the min value of this activation token, and can be viewed as the zero-point (Shao et al., 2023) of the quantizer.

As shown in Table 1, compared with the asymmetric uniform quantization (Appendix A.1) that encodes $T$ levels via $log_2T$ bits, IF neurons recursively encode spikes via total $T$ bits, because firing rate encoding directly makes an average of spiking steps, which missing the numerical carry. In ANN-SNN conversion (Bu et al., 2021; Li et al., 2021), IF neuron equals uniform quantization, where SNNs expand $T$ steps of their $log_2T$ bit quantized ANN counterpart. By summation over spiking steps, the IF neuron encodes the input into $log_2T$ bit integer values, where $\bar{\mathbf{s}}$ is the spike firing rate; Clip and Round indicate the min-max clipping and floor functions:

$$\mathbf{x}^{\text{INT}} = \text{Clip}\left(\text{Round}\left\lceil T\bar{\mathbf{s}}^{(\ell)}\right\rceil, 0, T\right). \tag{4}$$

### 3.2 LIMITATIONS OF TRADITIONAL QUANTIZATION

Traditional quantization is an ill-posed problem between bit-width and quantization error, especially for post-training LLMs. The performance drop is often caused by quantization errors of outliers (Xiao et al., 2023) and salient values (Dettmers et al., 2023). In details, the traditional LLM quantization faces the following challenges:

(i) weight-activation quantization makes it hard to avoid quantization errors in outliers. AWQ (Lin et al., 2023a) uses per-channel quantization step sizes to smooth outlier channels; however, it can only be applied to weight-only quantization, which is often insufficient for LLM compression.

(ii) Per-channel quantization is unfriendly to deployment. Since matrix multiplication is calculated per-token, per-channel quantization cannot be directly used to accelerate. As mitigation, SmoothQuant (Xiao et al., 2023) and OmniQuant (Shao et al., 2023) rebalance the quantization difficulty of activations and weights; however, they do not directly eliminate the impact of outliers.

(iii) Mix-precision quantization is hardware unfriendly. SpQR (Dettmers et al., 2023) and PB-LLM (Shang et al., 2023) use mixed-precision quantization to avoid quantizing salient weights; however, mix-precision introduces difficulties in hardware deployment.

Additionally, recent QuaRot (Ashkboos et al., 2024) achieves outlier free quantization by rotation. The Hadamard transformation in Quarot maps activations to roughly ball-shape Gaussian distribution, where more accurate spike encoding for larger values still benefits performance in our experiments. Moreover, compared with traditional quantization, QuaRot adds 4 Hadamard matrixes in each bloch, and their multiplications could not be absorbed. Based on these challenges, there is a requirement to explore spike encoding methods which helps accurate quantization.

## 4 SPIKE-DRIVEN QUANTIZATION

To avoid inefficient spike encoding in previous SNNs and significant quantization error in Quantized ANNs, we propose the first spiking large language model with two techniques: Generalized Integrate-and-Fire neurons for spike length comparison and Optimal Brain Spiking framework for accurate saliency-based spike encoding. Compared with IF-SNNs, SpikeLLM compresses binary code length from T to approximate $log_2 T$ and can infer with both the low-bit matrix multiplication workload on GPUs and the per-bit computation workload on neuromorphic chips.

### 4.1 GENERALIZED INTEGRATE-AND-FIRE NEURON

Because of the inefficiency of IF neurons, recent SNNs are almost less than 1 billion parameters and are hard to train with long-time backpropagation through time (BPTT). On the other hand, traditional quantization encodes all binary digits in one step, which makes it hard to quantize outliers. Based on both aspects, we make a balance between recursive steps and code length in each step. This is achieved by merging $L$ spiking steps and encoding each step as low-bit digits with $log_2 L$ bit length. We define this L-step merged IF neuron as the Generalized Integrate-and-Fire (GIF) neuron:

$$
\mathbf{s}_{GIF}(t') = \sum_{t=1}^{L} \mathbf{s}_{IF}(t), \ \mathbf{s}_{GIF}(t') = \begin{cases} k, & \text{if } kV_{th} \le L\mathbf{v}(t') < (k+1)V_{th}, k = 0, 1, ..., L-1 \\ L, & \text{if } \mathbf{v}(t) \ge V_{th} \end{cases},
$$

(5)

where there are L quantization levels in each spiking step. After merging, the recursive steps $T' = \frac{T}{L}, t' = \lfloor \frac{t}{L} \rfloor$ and each step has been encoded by $log_2 L$ bit quantization.

According to Eq.5, if simulating 32 ($T = 32$) quantization levels (or spiking steps), we can set the merged spiking step $T' = 2$ and spiking level $L = 16$ in each step. After merging, each step is quantized to $4 \times \{0, 1\}$ digits, and the total 32 quantization levels can be represented by $8 \times \{0, 1\}$ digits. Although GIF neuron still has longer code than traditional quantization, this gives us the freedom to choose spiking steps according to channel saliency in LLMs and achieve higher performance with similar costs.

**Remark 1.** Ternary Spike. *The ternary spike $\mathbf{s}_{Ter}(t)$ proposed by SpikeLM (Xing et al., 2024b) is a special case of GIF, which is formulated by merging a positive IF neuron $\mathbf{s}_{IF}^+(t)$ and a negative $\mathbf{s}_{IF}^-(t)$. Ternary spike not only increases quantization levels but also keeps additive in SNNs.*

$$
\mathbf{s}_{Ter}(t) = \mathbf{s}_{IF}^+(t) + \mathbf{s}_{IF}^-(t), \quad \mathbf{s}_{Ter}(t) = \begin{cases} -1, & \text{if } \mathbf{v}(t) < -V_{th} \\ 0, & \text{if } \mathbf{v}(t) \in (-V_{th}, +V_{th}) \\ +1, & \text{if } \mathbf{v}(t) > +V_{th} \end{cases}.
$$

(6)

**On-Chip Inference.** Due to the equivalence before and after spike merging in Eq. 5, we can choose different workflows for training and inference: (i) both multi-bit training and inference on GPUs: as shown in Figure 1 (d), GIF neurons can be converted to single-precision quantization and apply the general matrix multiplication kernels for inference, which equals to quantization but improves performance in Table 1. (ii) Multi-bit training on GPUs and 1-bit inference on neuromorphic chips: during training, merged spikes are employed, while during inference, the merged spikes are equivalent to expand back into their 1-bit formulation before merging. For example, if the value is $n$ in a step, it can be decoded as $n$ spikes occurred in $L$ time. This workflow avoids long-distance BPTT during training and therefore enhances training accuracy. We detial the training and inference of GIF neurons in Appendix A.4.

### 4.2 SALIENCY-AWARE SPIKING STEPS

Given the unequal importance of channels in LLMs, we propose a saliency-aware spiking mechanism to quantify this concept guiding the subsequent quantization process. This is addressed by expanding salient channels in activations or weights with more spiking steps compared with unimportant channels. Given the GIF neuron to quantize activations, we first detect salient channels $C'$ and the other channels $C$, and then allocate $T'$-step spikes to encode channels in $C'$ and one-step for

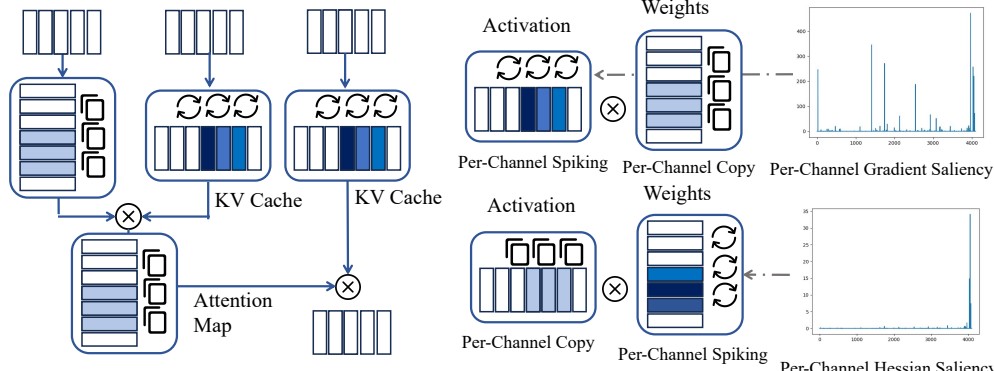

Figure 2: saliency-Aware spiking mechanisms in SpikeLLM. (Left) Spiking self-attention. Salient channels in the KV caches are encoded by multi-step spikes. (Right) Spiking activations or weights in a linear layer, where saliency is detected by gradient or Hessian metric respectively.

others in $C$, which can be represented by:

$$
\begin{aligned}
\mathbf{x}^{(\ell)} &= \frac{V_{th}}{T'} \sum_{t=1}^{T'} \mathbf{s}^{(\ell)}(t) + \min(\mathbf{x}^{(\ell-1)}) \\
&\simeq \frac{V_{th}}{T'} \sum_{t=1}^{T'} \mathbf{s}^{(\ell)}(t)|_{\mathbf{s} \in C'} + \mathbf{s}^{(\ell)}(1) V_{th}|_{\mathbf{s} \in C} + \min(\mathbf{x}^{(\ell-1)}),
\end{aligned}
\tag{7}
$$

where $V_{th} = \frac{\max(\mathbf{x})}{T'}$ is per-token spiking thresholds, which confirms not clipping max values (as Eq.4). As shown in Fig. 2, this per-channel spiking mechanism can apply to KV-caches, activations, and weights. For weight-activation quantization, salient channels in KV-caches and activations have $T'$ spiking steps, while the other side of the matrix multiplication keeps one-step quantization, and copies corresponding channels for the same $T'$ steps. For weight-only quantization, the salient channels in weights have $T'$ spiking steps, while the corresponding channels in activations are copied. Following, it is essential to detect the salient channels in both weights and activations.

### 4.3 OPTIMAL BRAIN SPIKING

Our Optimal Brain Spiking is a weight-activation generalization of the classic Optimal Brain Surgeon (OBS) framework (Hassibi & Stork, 1992). Different from OBS which focuses on weight pruning with only the second-order differentiation, we focus on detecting salient channels in both activations and weights via both first and second-order differentiation. Given a post-training model well-optimized under a loss function $\mathcal{L}$, any weights or activations $\mathbf{x}$ in the model can be expressed by a second-order Taylor expansion around its optimal value $\mathbf{x}^*$:

$$
\mathcal{L}(\mathbf{x}) \simeq \mathcal{L}(\mathbf{x}^*) + (\mathbf{x} - \mathbf{x}^*)^\top \nabla \mathcal{L}(\mathbf{x}^*) + \frac{1}{2}(\mathbf{x} - \mathbf{x}^*)^\top \mathbf{H}_{\mathcal{L}}(\mathbf{x}^*)(\mathbf{x} - \mathbf{x}^*),
\tag{8}
$$

where $\nabla \mathcal{L}(\mathbf{x}^*)$ and $\mathbf{H}_{\mathcal{L}}(\mathbf{x}^*)$ is the first-order differentiation and the second-order Hessian matrixes under the final loss $\mathcal{L}$ and we define $\delta \mathcal{L}(\delta \mathbf{x}) = \mathcal{L}(\mathbf{x}) - \mathcal{L}(\mathbf{x}^*)$. Specifically, for a linear layer with weights $\mathbf{W}$ and activations $\mathbf{X}$, we donate the quantization function as $\mathcal{Q}(.)$ and we have:

**Theorem 1.** Optimal Brain Spiking. *Given the layerwise objective to minimize the squared error, $argmin||\mathbf{W}\mathbf{X} - \mathcal{Q}(\mathbf{W})\mathcal{Q}(\mathbf{X})||_2^2$, the activation saliency is $\mathbf{X} \circ \mathbf{W}^\top \mathbf{W} \mathbf{X}$, and the weight saliency is $\frac{\mathbf{W}_{ij}^2}{[\mathbf{H}_{ii}^{-1}]^2}$, where the $\circ$ is Hadamard product.*

*Proof.* For activations, the gradient is not zero in a well-optimized model in Eq. 8, and we use the first-order Taylor expansion to approximate the effect of activation perturbations $\delta \mathbf{x}$: $\delta \mathcal{L}(\delta \mathbf{x}) \simeq \delta \mathbf{x}^\top \nabla \mathcal{L}(\mathbf{x}^*)$. Thus, the activation salient matrix is directly calculated according to $\delta \mathcal{L}(\delta \mathbf{x})$:

$$
\text{Saliency}(\mathbf{X}) = \mathbf{X} \circ \mathbf{W}^\top \frac{\partial \mathcal{L}}{\partial \mathbf{W} \mathbf{X}} = \mathbf{X} \circ \mathbf{W}^\top \mathbf{W} \mathbf{X}
\tag{9}
$$

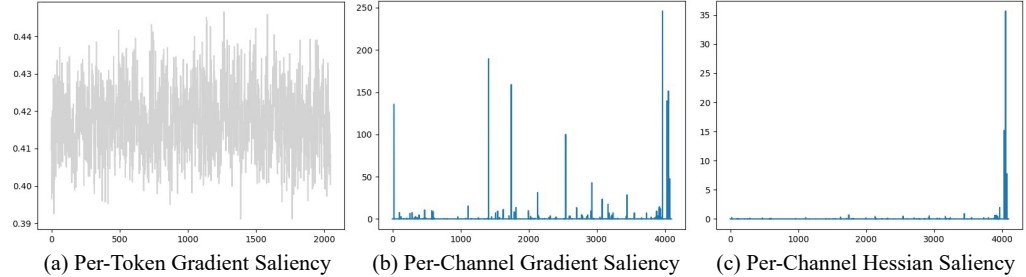

(a) Per-Token Gradient Saliency  (b) Per-Channel Gradient Saliency  (c) Per-Channel Hessian Saliency

Figure 3: Comparisons of different saliency metrics in the first linear layer. (a) Insignificant per-token gradient saliency in activations. (b) Significant per-channel gradient saliency in activations. (c) Significant per-channel Hessian saliency in weights. The horizontal axis represents each channel.

Table 2: Spiking settings to simulate quantization. We set 10% or 5% salient channels for 2 or 4 spiking steps in OBSpiking. Attention, Act. and Weight indicate where to apply OBSpiking.

| Quantizer | Spike-Level L | Step T' | Salient Channels | Attention | Act. | Weight |
|---|---|---|---|---|---|---|
| 4bit | 16 | 2 | 10% | ✓ | ✓ | – |
| 4bit | 16 | 4 | 5% | ✓ | ✓ | – |
| 2bit | 4 | 2 | 10% | – | – | ✓ |
| 2bit | 4 | 4 | 5% | – | – | ✓ |

For weights, the gradient is zero because the optimizer directly optimizes weights to the local minimum after pretraining. Thus, the first-order term is zero in Eq.8 and $\delta\mathcal{L}(\delta\mathbf{w})$ has to approximate via the second-order term in Taylor expansion: $\delta\mathcal{L}(\delta\mathbf{w}) \simeq \frac{1}{2}\delta\mathbf{w}^\top \mathbf{H}_\mathcal{L}(\mathbf{w}^*)\delta\mathbf{w}$, which is proved in OBS (Hassibi & Stork, 1992). And we apply the same weight saliency metric as OBS-based methods (Shang et al., 2023; Frantar et al., 2022; Frantar & Alistarh, 2022):

$$\text{Saliency}(\mathbf{W}_{ij}) = \frac{\mathbf{W}_{ij}^2}{[\mathbf{H}_{ii}^{-1}]^2}, \quad \mathbf{H} = 2\mathbf{X}\mathbf{X}^\top. \tag{10}$$

$\square$

**Remark 2.** Per-Channel Spiking Mechanism. *Given saliency matrixes Saliency($\mathbf{X}$) and Saliency($\mathbf{W}$) from Optimal Brain Spiking, per-channel means of first- (or second-) order differentation-based saliency are significant enough to divide salient channels in Eq.7.*

In implmentation, the saliency matrix Saliency($\mathbf{X}$) and Saliency($\mathbf{W}$) have the same shape with activations $\mathbf{X}$ and weights $\mathbf{W}$, which are inefficient to store. As shown in Fig.3, we calculate per-channel or per-token means of the saliency matrix. We observe that both the per-channel saliency in activations and weights are robust enough to detect salient channels while per-token is insignificant. Based on *Remark 2*, we first compute per-channel saliency with calibration data and generate their rank to select the salient channels in Eq.7. Then, we store these lightweight masks for inference.

## 5 EXPERIMENTS

We evaluate the necessity to introduce SpikeLLM due to the inefficiency and incompatibility caused by the direct application of existing ANNs based quantisation strategies to spike neural networks. This necessity is studied and verified in two main directions: (i) general weight-activation quantization in very low bits; (ii) ternary quantized LLM towards additive linear layers.

**Training.** As shown in Table 2, SpikeLLM can simulate W4A4 (4-bit activation, 4-bit weight), W2A8, or W2A16 quantization with different hyper-parameters, including the spiking step $T'$ and spike-level $L$ in GIF neurons, and the ratio of saliency channels in OBSpiking. We follow main streams of post-training quantization (PTQ) (Frantar et al., 2022) and calibration (Shao et al., 2023) pipelines. (i) For the low-bit quantization setting, our primary baseline is OmniQuant (Shao et al., 2023) and we report the detailed pipeline in Appendix A.2. We select LLAMA-1-7B (Touvron et al., 2023a) and LLAMA-2-7B, 13B, 70B (Touvron et al., 2023b) as full-precision models. (ii) For the additive SpikeLLM setting, we follow the GPTQ (Frantar et al., 2022) framework and report training details in Appendix A.3. Both pipelines use the same 128 WikeText2 calibration data for continue

Table 3: Comparisons with quantized LLMs. SpikeLLM$_T$ is defined in Table 2, where T is the spiking step in saliency channels, the same as T' in Table 2. OmniQuant† is trained with the unified settings in Appendix A.2. Results are evaluated by the old version lm-eval, the same as Omniquant.

| Method | Saliency | #Bits | ACEs | PIQA$_{norm}$ | ARC-e$_{norm}$ | Arc-c$_{norm}$ | BoolQ | HellaSwag$_{norm}$ | Winogrande | Avg. |
|---|---|---|---|---|---|---|---|---|---|---|
| **LLAMA-1-7B** | – | FP16 | 1× | 77.37 | 52.48 | 41.38 | 73.12 | 72.99 | 66.93 | 64.05 |
| SmoothQuant | – | W4A4 | 0.0625× | 49.80 | 30.40 | 25.80 | 49.10 | 27.40 | 48.00 | 38.41 |
| LLM-QAT | – | W4A4 | 0.0625× | 51.50 | 27.90 | 23.90 | 61.30 | 31.10 | 51.90 | 41.27 |
| LLM-QAT+SQ | – | W4A4 | 0.0625× | 55.90 | 35.50 | 26.40 | 62.40 | 47.80 | 50.60 | 46.43 |
| OS+ | – | W4A4 | 0.0625× | 62.73 | 39.98 | 30.29 | 60.21 | 44.39 | 52.96 | 48.43 |
| OmniQuant† | – | W4A4 | 0.0625× | 63.22 | 40.99 | 29.86 | 62.23 | 50.77 | 52.49 | 49.93 |
| **SpikeLLM$_{T=2}$** | 0.10 | W4A4 | 0.0688× | 65.45 | 41.67 | 32.51 | 64.37 | 56.59 | 54.3 | **52.48** |
| **LLAMA-2-7B** | – | FP16 | 1× | 76.99 | 53.58 | 40.61 | 71.07 | 72.96 | 67.25 | 63.74 |
| OmniQuant | – | W4A4 | 0.0625× | 61.75 | 40.32 | 29.86 | 60.89 | 48.86 | 52.17 | 48.98 |
| **SpikeLLM$_{T=2}$** | 0.10 | W4A4 | 0.0688× | 62.35 | 41.41 | 29.95 | 58.87 | 54.27 | 50.20 | **49.51** |
| OmniQuant | – | W2A8 | 0.0625× | 50.71 | 27.74 | 25.17 | 38.26 | 26.16 | 50.36 | 36.40 |
| **SpikeLLM$_{T=4}$** | 0.05 | W2A8 | 0.075× | 53.37 | 31.27 | 23.63 | 53.79 | 33.94 | 51.46 | **41.24** |
| OmniQuant | – | W2A16 | 0.125× | 57.02 | 32.83 | 25.34 | 53.46 | 31.50 | 50.36 | 41.75 |
| **SpikeLLM$_{T=2}$** | 0.10 | W2A16 | 0.138× | 65.67 | 41.88 | 28.41 | 60.46 | 49.87 | 52.80 | **49.85** |
| **LLAMA-2-13B** | – | FP16 | 1× | 79.05 | 57.95 | 44.28 | 69.02 | 76.63 | 69.61 | 66.09 |
| OmniQuant | – | W4A4 | 0.0625× | 68.01 | 46.09 | 32.51 | 62.87 | 57.71 | 53.91 | 53.52 |
| **SpikeLLM$_{T=2}$** | 0.10 | W4A4 | 0.0688× | 66.0 | 45.71 | 33.19 | 64.07 | 61.38 | 54.54 | **54.15** |
| OmniQuant | – | W2A8 | 0.0625× | 56.96 | 33.0 | 23.55 | 61.31 | 32.0 | 49.41 | 42.71 |
| **SpikeLLM$_{T=4}$** | 0.05 | W2A8 | 0.075× | 64.69 | 41.46 | 28.16 | 62.29 | 50.5 | 52.49 | **49.93** |
| OmniQuant | – | W2A16 | 0.125× | 62.24 | 40.15 | 29.52 | 62.45 | 49.76 | 53.12 | 49.54 |
| **SpikeLLM$_{T=2}$** | 0.10 | W2A16 | 0.138× | 67.68 | 44.53 | 30.38 | 63.64 | 58.02 | 57.22 | **53.58** |
| **LLAMA-2-70B** | – | FP16 | 1× | 80.85 | 59.72 | 47.95 | 76.7 | 80.85 | 77.03 | 70.52 |
| OmniQuant | – | W2A16 | 0.125× | 68.44 | 45.16 | 32.59 | 63.12 | 58.32 | 52.25 | 53.31 |
| **SpikeLLM$_{T=2}$** | 0.10 | W2A16 | 0.138× | 75.84 | 51.68 | 39.42 | 66.67 | 68.44 | 60.22 | **60.38** |

Table 4: Comparisons with QuaRot. We follow evaluation metrics of QuaRot and all the results are evaluated by lm-eval-v0.4.2 following recent works. Saliency indicates the ratio of salient channels, and we set spiking step, T'=2, in salient channels.

| Method | Saliency | #Bits | PIQA$_{norm}$ | ARC-e$_{norm}$ | ARC-c$_{norm}$ | BoolQ | HellaSwag$_{norm}$ | Winogrande | Avg. |
|---|---|---|---|---|---|---|---|---|---|
| LLAMA-2-7B | - | FP16 | 78.84 | 74.54 | 46.33 | 77.74 | 75.97 | 69.22 | 70.44 |
| QuaRot-RTN | - | W4A4 | 71.82 | 59.89 | 36.18 | 67.37 | 63.88 | 59.12 | 59.71 |
| SpikeLLM-RTN | 0.2 | W4A4 | 72.47 | 62.29 | 36.01 | 69.48 | 64.74 | 59.43 | **60.74** |
| QuaRot-GPTQ | - | W4A4 | 75.95 | 68.43 | 39.76 | 72.54 | 72.23 | 64.72 | 65.61 |
| SpikeLLM-GPTQ | 0.2 | W4A4 | 77.37 | 70.88 | 41.81 | 73.00 | 72.42 | 65.11 | **66.76** |
| LLAMA-2-13B | - | FP16 | 80.63 | 77.48 | 49.23 | 80.73 | 79.37 | 71.74 | 80.69 |
| QuaRot-RTN | - | W4A4 | 74.86 | 69.19 | 41.98 | 72.54 | 70.35 | 64.72 | 65.61 |
| SpikeLLM-RTN | 0.2 | W4A4 | 75.79 | 69.53 | 41.21 | 74.31 | 71.51 | 65.51 | **66.31** |
| QuaRot-GPTQ | - | W4A4 | 77.91 | 72.18 | 46.16 | 78.41 | 75.55 | 68.82 | 69.84 |
| SpikeLLM-GPTQ | 0.3 | W4A4 | 78.45 | 73.74 | 46.84 | 78.75 | 76.40 | 68.03 | **70.37** |

layer-wise training based on full-precision LLMs. It takes around 3.5 hours to train a 7B model with OmniQuant pipeline on a single A100-80 GPU.

**Evaluation.** We follow the same evaluation methods as the primary baselines, OmniQuant (Shao et al., 2023) and PB-LLM (Shang et al., 2023). We evaluate the perplexity (PPL) of language generation in WikiText2 (Merity et al., 2016) and C4 (Raffel et al., 2020) benchmarks. We also evaluate zero-shot common scene reasoning tasks including PIQA (Bisk et al., 2020), ARC-easy (Clark et al., 2018), ARC-challenge (Clark et al., 2018), BoolQ (Clark et al., 2019), HellaSwag (Clark et al., 2018), and Winogrande (Sakaguchi et al., 2021) datasets.

**Operation Metric.** (i) For GEMM kernels, we make a direct comparison of SpikeLLM and quantized ANNs by introducing the ACE metric (Zhang et al., 2022b) (Appendix A.1), which is defined as the number of binary operations, $ACE = MACs \times bit_{weight} \times bit_{act.}$, where $bit_{weight}$ and $bit_{act.}$ indicates the bit-width of weights and activations respectively. (ii) For inference on neuromorphic chips, we consider the event-driven sparsity in SNNs. We introduce a Sparse ACE, defined as $sparsity \times ACE$, which is evaluated after expanding GIF neurons back to 1-bit spiking.

## 5.1 MAIN RESULTS

**Comparison with General Quantization.** As shown in Table 3 and A.5.1, we compare SpikeLLM with the most general weight-activation quantization methods including SmoothQuant, LLM-QAT, and OmniQuant, which shows SpikeLLM improves significantly based on OmniQuant with a few additional spikes to enhance salient channels. When replacing activation and KV-cache quantiza-

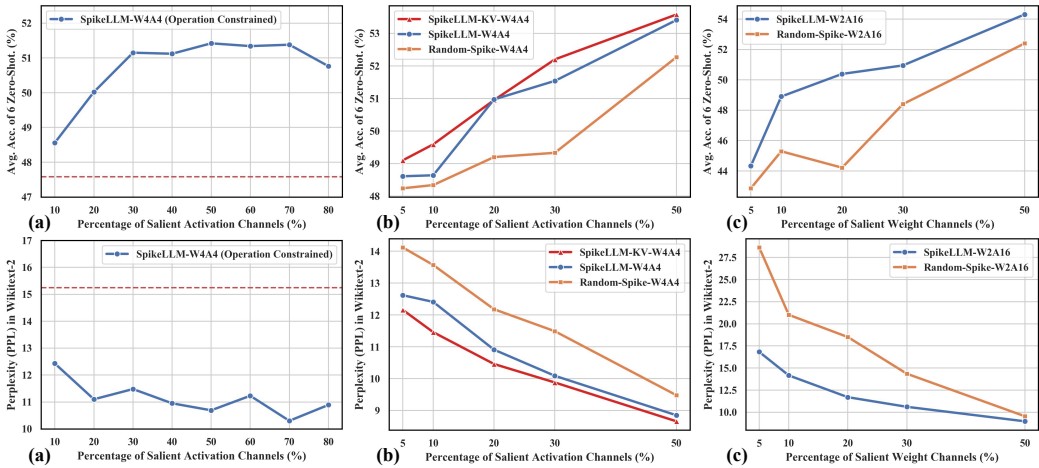

Figure 4: Ablation studies in LLAMA-2-7B. Average accuracy (not norm) is reported. (a) Comparison between SpikeLLM and Quantized-ANN with the same operations. (b) Ablations on spiking salient channels in activations and KV-Caches. (c) Ablations on spiking salient channels in weights.

tion with spiking neurons, SpikeLLM-W4A4 LLAMA-7B improves 2.55% accuracy and reduces 11.01% perplexity. When replacing weight quantization, improvements is more dramatic: for the LLAMA-2-7B-W2A8 and W2A16, SpikeLLM$_{T=4}$ and SpikeLLM$_{T=2}$ exceed 4.84% and 8.10% respectively; 89.58% and 61.89% perplexity are reduced. Notice that, SpikeLLM can plug in almost quantized LLMs beyond OmniQuant, which indicates the potential higher performance equipped with stronger quantization pipeline. In Appendix A.5, we further explore more training pipelines, such as QLoRA (Dettmers et al., 2024) and efficient QAT (Chen et al., 2024) settings.

**Comparison with QuaRot.** Recent QuaRot (Ashkboos et al., 2024) achieves outlier-free quantization with the help of additional Hadamard transformations. After rotation, the activation become a roughly ball-shape Gaussian distribution. In SpikeLLM framework, we select the more salient channels given a Gaussian distribution, and allocate more steps. As shown in Table 4, SpikeLLM improves 1.03% and 1.15% when quantizing weights by RTN and GPTQ respectively in LLAMA-2-7B, indicating the ability to cooperate with rotation-based quantization.

**Operation Efficiency Compared with ANNs.** In Table 5, we evaluate the multi-bit and 1-bit workflows proposed in Sec. 4.1. Compared to LLAMA-7B, SpikeLLM$_{T=2}$ for the W4A4 setting saves ×10.79 ACEs or ×6.38 sparse ACEs respectively. Because GIF neurons expand much longer in 1-bit inference, it achieves asynchronous computation for the following linear layer at the cost of more operations than GEMM kernels.

Table 5: Operations of SpikeLLM. ACE and Sparse ACE evaluate operations on the GEMM kernel and event-driven neuromorphic chips.

| Models | ACE ($10^{12}$) | Sparse ACE ($10^{12}$) |
|---|---|---|
| LLAMA-1-7B | 3886.22 | 3886.22 |
| SpikeLLM$_{T=2}$ | 360.20 | 608.78 |
| SpikeLLM$_{T=4}$ | 386.70 | 648.09 |

**Ablation Studies on GIF neurons.** We evaluate the efficiency of GIF neurons by comparison with OmniQuant-W4A4 with almost the same operations. This is achieved by applying GIF neurons in both weights and activations. It is known that GIF neurons and OBSpiking always introduce a few additional operations. To confirm the same operations, we accordingly decrease the spike length in non-salient channels in weights and maintain the total spike length of weights and activations the same as OmniQuant-W4A4. As shown in Fig.4 (a), given different percentages of salient channels in activations, SpikeLLM always exceeds the Omniquant-W4A4 baseline shown as the red line.

**Ablation Studies on Optimal Brain Spiking.** To evaluate the efficiency of the OBSpiking framework, we compare SpikeLLM with equal-operation baselines with randomly selected spiking channels, termed Random-Spike. In Fig. 4 (b), we compare three settings: (i) SpikeLLM-W4A4: OBSpiking in activations of linear layers alone, (ii) SpikeLLM-KV-W4A4: OBSpiking in both activations of linear layers and KV caches of self-attentions, and (iii) Random-Spike in activations of linear layers alone. We don't evaluate the random-spike in KV-caches because this setting is unstable to optimize. With different percentages of salient channels, OBSpiking in both KV chches and

Table 6: Comparisons between SpikeLLM and PB-LLM in LLAMA-2-7B towards additive linear layers. SpikeLLM$_{Ter,x:y:z}$ indicates using the ternary GIF neurons in Eq.6 as weight quantizers, where x:y:z is percentages of 1, 2, 4 spiking steps in weights (details in Appendix A.3).

| Method | ACs | Equal Steps | PIQA | ARC-e | Arc-c | BoolQ | HellaSwag | Winogrande | **Avg.** |
|---|---|---|---|---|---|---|---|---|---|
| PB-LLM$_{80\%}$ | 80% | 2.4 | 60.77 | 43.9 | 22.18 | 64.16 | 33.75 | 56.83 | 46.93 |
| PB-LLM$_{90\%}$ | 90% | 1.7 | 54.03 | 27.9 | 19.37 | 57.09 | 27.12 | 48.38 | 38.98 |
| PB-LLM$_{95\%}$ | 95% | 1.35 | 53.43 | 26.6 | 19.28 | 51.87 | 26.51 | 49.01 | 37.78 |
| SpikeLLM$_{Ter,70:25:5}$ | 100% | 1.4 | 65.83 | 51.89 | 25.17 | 68.47 | 40.48 | 60.77 | 52.10 |
| SpikeLLM$_{Ter,80:15:5}$ | 100% | 1.3 | 60.88 | 42.26 | 24.23 | 68.65 | 34.02 | 54.38 | 47.40 |
| SpikeLLM$_{Ter,85:10:5}$ | 100% | 1.25 | 55.44 | 31.99 | 20.99 | 61.83 | 30.02 | 52.01 | 42.05 |
| SpikeLLM$_{Ter,90:5:5}$ | 100% | 1.2 | 53.75 | 28.83 | 19.2 | 37.92 | 28.46 | 48.38 | 36.09 |

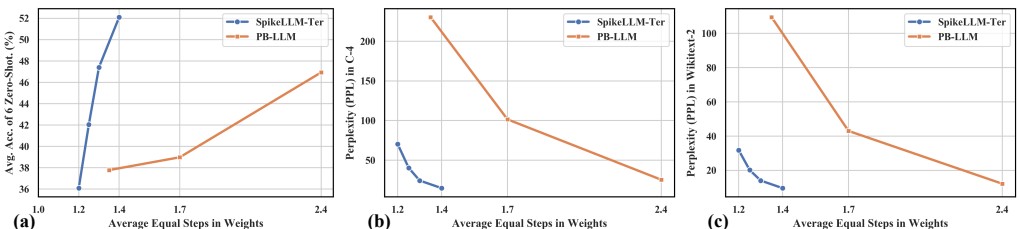

Figure 5: Effiency comparisons of SpikeLLM$_{Ter}$ and PB-LLM in Wikitext-2, C4 and 6 zero-shot benchmarks. We use the average of equal steps as the operation metric of SNNs and BNNs.

activations helps to improve performance. In Fig. 4 (c), we compare Random-Spike and OBSpiking in weight quantization, which proves the effectiveness for weights similarly.

## 5.2 ADDITIVE SPIKING LLMS

To direct maintain the addition nature and event-driven sparsity of SNNs without expanding GIF neurons, we additionally build SpikeLLM$_{Ter}$ based on the ternary GIF neuron in Eq.6 as the spiking weight quantizer. After weight quantization as Eq.6, linear layers can be implemented by ACcumulation (AC). The detailed training and evaluation settings are reported in Appendix A.3.

**Comparison with binary LLM.** We select PB-LLM for comparison since binary weight neural networks (BNNs) can be implemented by ACs. As shown in Table 6, SpikeLLM achieves full AC operations in linear layers by saliency-based spiking neuronal dynamics, instead of the deployment-unfriendly mixed-precision quantization in PB-LLM.

**Efficiency of additive SpikeLLM.** In Table 6, we evaluate the AC operations in a linear layer by equal-steps. For PB-LLM, we view the average bit-width as the equal steps; for SpikeLLM, the equal steps are calculated by the average spiking steps of salient and other values. As shown in Table 6, SpikeLLM is able to exceed PB-LLM with similar equal-steps. Further, in Fig 5, we evaluate the Pareto front about equal-steps and performance, which shows SpikeLLM exceeds PB-LLM (BNN) in both effectiveness and efficiency.

## 6 CONCLUSION

We propose the first spiking large language models with 7∼70 billion parameters, promoting SNNs to the era of LLMs. This is achieved by significant improvement of spike encoding efficiency, where the GIF neurons compress the code length from T to $\frac{T}{L}log_2L$ bits; the OBSpiking further compresses to approximate $log_2T$ bits. Unlike previous ANN-SNN conversions that rely on quantization, this work exceeds quantization with the hybrid encoding of both SNNs and quantized-ANNs. Scaling up spiking neuronal dynamics in LLMs and developing training efficient spiking mechanisms for tens of billion parameters have become increasingly valuable issues.

**Limitations** A primary limitation of this paper is the performance gap between Spiking LLMs and ANN LLMs. This work applies ANN-SNN calibration with only 128 calibration data and finishes training a 7B model in 3.5 GPU hours. Continuous pretraining would boost the performance.

ACKNOWLEDGMENTS

This work is supported by National Science and Technology Major Project (2023ZD0121504). This work is partially supported by CAS Project for Young Scientists in Basic Research (YSBR-116), National Natural Science Foundation of China (62325603, 62236009), Beijing Science and Technology Plan (Z241100004224011).

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

# A  APPENDIX

## A.1  LOW-BIT QUANTIZATION

Low-bit quantization typically maps full-precision value to fewer quantization levels. We first focus on the most widely used asymmetric uniform quantization. Given the full-precision value $\mathbf{x}$, the quantizer first maps $\mathbf{x}$ to a `INT` value by linear translation and `Round` functions, and then maps the discrete `INT` value back to its original range, which the former translation can be expressed as:

$$\mathbf{x}^{\text{INT}} = \text{Round}\lceil\frac{\mathbf{x}^{\text{FP16}} - \min(\mathbf{x}^{\text{FP16}})}{\Delta}\rfloor, \quad \Delta = \frac{\max(\mathbf{x}) - \min(\mathbf{x})}{2^N - 1}. \tag{11}$$

In linear layers, quantized weights $w^q$ or activations $a^q$ can be represented as binary digits in M or N bits: $a^q = \sum_{i=0}^{M-1} \mathbf{a}_i 2^i$, $w^q = \sum_{j=0}^{N-1} \mathbf{w}_j 2^j$. Therefore, the Multiply-ACcumulate (MAC) can be implemented by bit level `AND` and `PopCount` operations:

$$a^q \cdot w^q = \sum_{i=0}^{M-1} \sum_{j=0}^{N-1} 2^{i+j} \text{PopCount}[\text{AND}(\mathbf{a}_i, \mathbf{w}_j)]. \tag{12}$$

This indicates the complexity and energy consumption of MAC operations are proportional to $M \times N$. Therefore, the number of operations in a quantized model can be measured using the arithmetic computation effort (ACE) metric (Zhang et al., 2022b), which is defined as $M \times N$ for a MAC operation between the M-bit weight and N-bit activation. Recent LLMs contain more than $10B \sim 100B$ parameters, making it an essential requirement to push not only weights but also activations to lower bit-width.

## A.2  TRAINING DETAILS OF THE OMNIQUANT PIPELINE.

### A.2.1  TRAINING DETAILS.

We use a unified training config as shown in all of our experiments. We train LLAMA-1 (7B), and LLAMA-2 (7B, 13B, 70B) for both OmniQuant baselines and SpikeLLMs according to this training scheme. Compared with the original OmniQuant, we do not apply loss augmentation methods except for 70B models for training stability and we set the quantization group number as 1 in all of the experiments. Therefore, this scheme can be viewed as a simplified version without bells and whistles to focus on the influence of the quantization method itself.

In SpikeLLM, we compute the saliency metric for activations layer by layer during the OmniQuant pipeline. Different from activations, we compute the saliency metric for weights direct using the features from the first embedding layer. Because we find this can make the computation of the inverse Hessian matrix more stable compared with computing layer by layer.

Table 7: Training settings on the OmniQuant scheme. LET and LWC indicate learnable equivalent transformation and learnable weight clipping.

| config | 4W4A | 2W8A | 2W16A |
|---|---|---|---|
| LET | True | True | False |
| LWC | True | True | True |
| learning rate of LET | 0.001 | 0.001 | N/A |
| learning rate of LWC | 0.01 | 0.01 | 0.01 |
| activation smooth | 0.75 | N/A | N/A |
| batch size | 1 | 1 | 1 |
| loss augmentation | False | False | True |
| epochs | 20 | 20 | 40 |
| group | 1 | 1 | 1 |

### A.2.2  TRAINING DATA.

Following OmniQuant (Shao et al., 2023), we randomly select 128 calibration data from WikiText2, which are 2048-token chunks. We further investigate the influence of different training samples to confirm the robustness of the proposed methods. In the OmniQuant pipeline, training samples are randomly cropped from the Wikitext2 dataset. To compare the performance of SpikeLLM and the

OmniQuant baseline, we use a set of random seeds to sample different training data each time. In Fig. 6, we sample data and train 16 times, using the same seed for both OmniQuant and SpikeLLM each time. SpikeLLM achieves higher average accuracy on 6 zero-shot datasets and lower Wikitext2 or C4 perplexity at the same time, showing consistently better performance. For the same training data, SpikeLLM always performs better. In other experiments, we keep the random seed as 2.

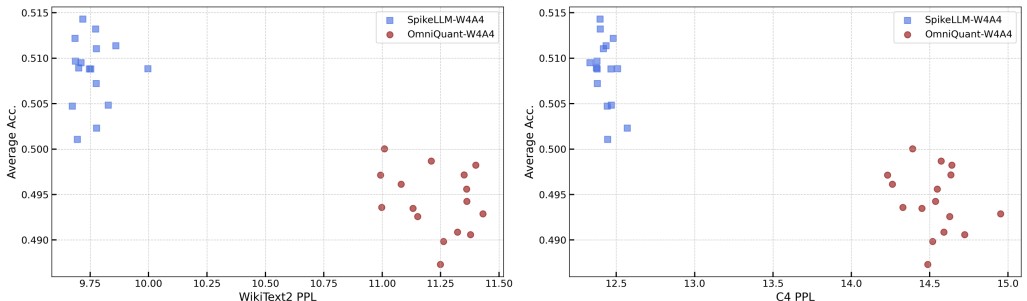

Figure 6: Comparison of different training data for LLAMA-1-7b W4A4 models. In SpikeLLMs, 10% activation channels are set to 2 spiking steps. Loss augmentations are applied as OmniQuant (Shao et al., 2023). The average accuracy (not norm) is evaluated.

## A.3 TRAINING DETAILS OF THE TERNARY SPIKELLM.

### A.3.1 MODEL DEFINITION.

In additive SpikeLLM, we apply the ternary spike level $\{-1,0,+1\}$ to encode weights according to Eq. 6. After weight ternarization, the matrix multiplication in the linear layer can be implemented by full ACcumulation (AC). As shown in Table 8, we have 3 spiking-step settings. For example, for the SpikeLLM$_{\text{Ter},70:25:5}$ model, we allocate the most salient 5% values in weights with 4 spiking steps, the following 25% with 2 spiking steps, and the rest 70% with 1 spiking steps. The unstructured spiking steps are applied, which is different from the experiment in the Main Result Section. As the following section, this is because of more accurate quantization in ultra-low bit quantization and fair comparison with PB-LLM (Shang et al., 2023).

Table 8: Model settings in additive SpikeLLM. 1,2,4-Steps indicate the percentages of 1, 2, and 4 spiking-step encoding in weights respectively.

| Model | 1-Step | 2-Step | 4-Step | Avg. Step | Spike-Level |
|---|---|---|---|---|---|
| SpikeLLM$_{\text{Ter},70:25:5}$ | 70% | 25% | 5% | 1.4 | $\{-1,0,+1\}$ |
| SpikeLLM$_{\text{Ter},80:15:5}$ | 80% | 15% | 5% | 1.3 | $\{-1,0,+1\}$ |
| SpikeLLM$_{\text{Ter},85:10:5}$ | 85% | 10% | 5% | 1.25 | $\{-1,0,+1\}$ |
| SpikeLLM$_{\text{Ter},90:5:5}$ | 90% | 5% | 5% | 1.2 | $\{-1,0,+1\}$ |

### A.3.2 STRUCTURED VS. UNSTRUCTURED SPIKING IN WEIGHTS.

For weight quantization, as shown in Table 9, unstructured spiking steps usually achieve higher performance compared with structured ones. Our per-channel spiking scheme can achieve higher performance by setting per-channel spiking steps as elementwise spiking steps. However, unstructured conditions need additional masks and are less friendly to deployment. Therefore, we keep structured settings in low-bit quantization. But for additive LLMs, the performance is more important in the extreme case, and we choose the unstructured settings. Moreover, the PB-LLM baselines are also unstructured, so that, it can also confirm the fair comparison.

Table 9: Comaprison between structured and unstructured weight quantization in the OmniQuant pipeline.

| Method | Saliency | #Bits | ACEs | PIQA | ARC-e | Arc-c | BoolQ | HellaSwag | Winogrande | Avg. |
|---|---|---|---|---|---|---|---|---|---|---|
| **LLAMA-2-7B** | – | FP16 | 1× | 78.45 | 69.32 | 40.02 | 71.07 | 56.69 | 67.25 | 63.80 |
| OmniQuant | – | W2A16 | 0.125× | 57.13 | 35.02 | 21.16 | 53.46 | 29.32 | 50.36 | 41.08 |
| SpikeLLM$_{\text{T}=2}$-Structured | 0.10 | W2A16 | 0.138× | 65.61 | 48.15 | 27.39 | 60.46 | 39.01 | 52.80 | 48.90 |
| SpikeLLM$_{\text{T}=2}$-Unstructured | 0.10 | W2A16 | 0.138× | 72.63 | 60.06 | 30.89 | 65.05 | 48.52 | 59.51 | 56.11 |

### A.3.3 TRAINING DETAILS.

The same as PB-LLM (Shang et al., 2023), we follow the GPT-Q (Frantar et al., 2022) pipeline for the post-training weight quantization. In quantization, we keep the same block size of 128. We apply the same 128 calibration data as PB-LLMs. Each data is randomly selected from WikiText2 and tokenized to formulate a 2048-token chunk.

## A.4 SPIKING NEURONAL DYNAMICS

### A.4.1 STEP MERGING IN GIF NEURONS

In Fig. 7, we illustrate the function Eq. (5). After merging multiple steps (L) of the IF neuron, the multi-step spike become a floor function, which can activate multi-level firing. The membrane potential of the GIF neuron will fire according to Eq. (5). Finally, the output of GIF neuron is according to Eq. (3) : $\mathbf{x}^{(\ell-1)}(t) = \mathbf{s}^{(\ell)}(t)V_{th} + \min(\mathbf{x}^{(\ell-1)})$.

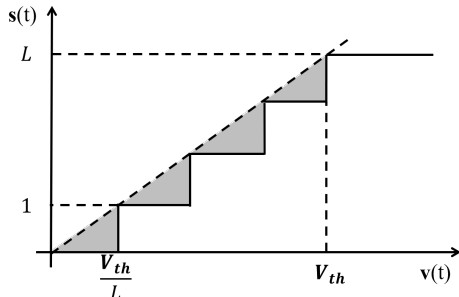

Figure 7: The floor firing function of the GIF neuron.

### A.4.2 COMPARISON OF INFERENCE

(i) Neuronal Dynamics of IF Neurons (Fig. 8 (a)): Given a real-valued input sequence, the membrane potential accumulates over time. When it exceeds the firing threshold, the IF neuron emits a spike, and the membrane potential is reset by subtracting the threshold value.

(ii) Neuronal Dynamics of Multi-Step GIF Neurons (Fig. 8 (b)): Similar to IF neurons, GIF neurons accumulate membrane potential over time steps. However, unlike IF neurons, GIF neurons first integrate inputs over a fixed number of steps L. Based on Eq. (5), GIF neurons can fire k discrete levels of spikes, where $k \in [0, L]$.

(iii) Neuronal Dynamics of Binary GIF Neurons (Fig. 8 (c)): The only difference between multi-bit and binary inference of GIF neurons lies in the firing threshold. GIF neurons in multi-bit inference have multiple thresholds corresponding to different firing levels, while binary inference uses a single threshold, similar to IF neurons. As a result, binary inference emits one bit of spike per time step, as a result, multi-level spikes are effectively decoded as binary. Next, we demonstrate the equivalence of binary inferenced GIF neurons and IF neurons.

Equivalence Conditions for Spiking Neurons:

(i) Given the same real-valued input sequence with firing rate encoding, the neurons are considered equivalent if the spike firing rates are identical.

(ii) For finite-step spikes, the membrane potential at the end of a given period must also be the same to ensure consistent firing rates in subsequent steps.

Based on these conditions, since the binary GIF and IF neurons share the same inputs and firing threshold over a short time window, their firing rates and membrane potential values are identical, confirming their equivalence in binary inference.

### A.4.3 COMPARISON OF TRAINING

As shown in Fig. 9, we compare the training dynamics of conventional spiking neural networks (SNNs) and GIF neurons. The discussions are as follows.

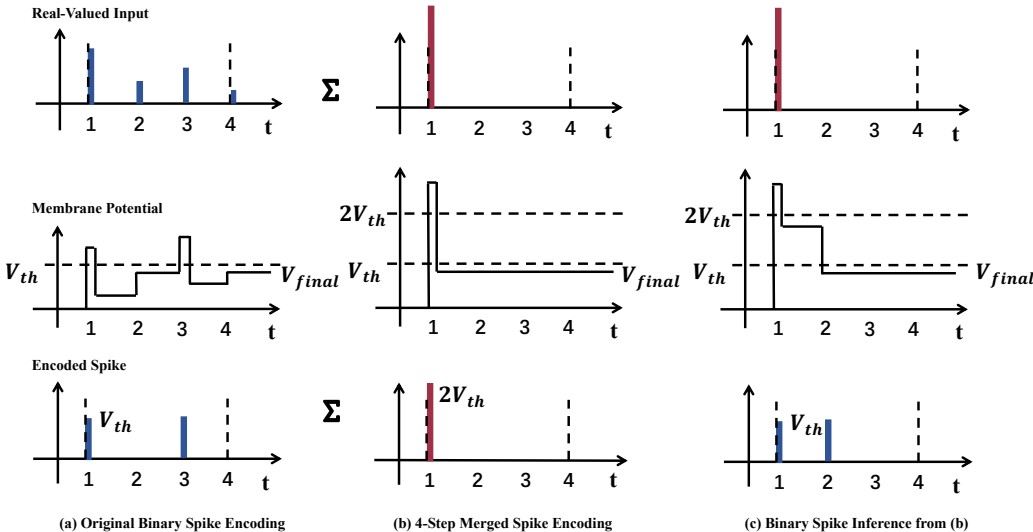

Figure 8: Neuronal dynamics of GIF neurons. We compare the neuron input, membrane potential, and output spike in three conditions: (a) the traditional IF neuron. (b) Multi-step merged GIF neuron. (c) Binary inference of GIF neuron.

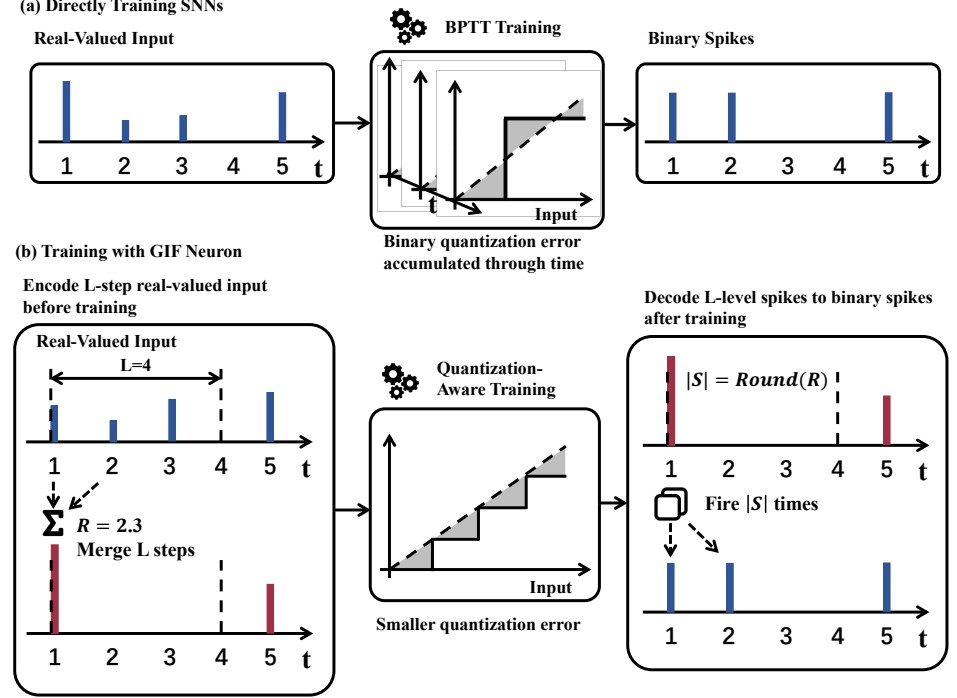

Figure 9: Training dynamics of GIF neurons. We compare the traditional backpropagation through time (BPTT), quantization-aware training in GIF neurons.

(i) Fig. 9 (a): Traditional backpropagation through time (BPTT) for IF (or LIF) neurons suffers from significant quantization errors due to binary quantization at each time step. These errors accumulate over time, making SNNs difficult to train.

(ii) Fig. 9 (b): GIF neurons use quantization-aware training. By aggregating L time steps, GIF neurons perform multi-bit quantization, which eliminates inter-step error accumulation. The multi-bit quantization improves gradient estimation (STE), leading to more accurate training.

## A.5 MORE RESULTS

### A.5.1 PERPLEXITY IN OMNIQUANT SETTINGS

Table 10: Comparisons between SpikeLLM and OmniQuant in the Wikitext2 and C4 PPL metrics. The LLAMA-2 family is evaluated. We do not evaluate the W4A4 and W2A8 settings for LLAMA-2-70B because the grouped-query attention (GQA) makes training unstable in the Omni-Quant pipeline.

| Method | Saliency | #Bits | LLAMA-2-7B | | LLAMA-2-13B | | LLAMA-2-70B | |
|---|---|---|---|---|---|---|---|---|
| | | | Wikitext2 | C4 | Wikitext2 | C4 | Wikitext2 | C4 |
| OmniQuant | – | W4A4 | 15.25 | 19.35 | 12.40 | 15.87 | – | – |
| SpikeLLM$_{\tau=2}$ | 0.10 | W4A4 | 11.93 | 15.34 | **9.64** | **12.48** | – | – |
| SpikeLLM$_{\tau=4}$ | 0.05 | W4A4 | **11.85** | **15.18** | 12.17 | 15.71 | – | – |
| OmniQuant | – | W2A8 | 287.64 | 445.21 | 53.87 | 72.33 | – | – |
| SpikeLLM$_{\tau=4}$ | 0.05 | W2A8 | **29.98** | **46.61** | **12.80** | **17.06** | – | – |
| OmniQuant | – | W2A16 | 38.05 | 98.74 | 17.14 | 27.12 | 10.04 | 19.31 |
| SpikeLLM$_{\tau=2}$ | 0.10 | W2A16 | **14.16** | **19.73** | **9.45** | **13.86** | **6.12** | **8.82** |

### A.5.2 COMPARISON WITH QLoRA METHODS

This line of work focuses on adding a low-rank branch to finetune the low-bit quantized model. State-of-the-art methods including IR-QLoRA (Qin et al., 2024), QA-LoRA (Xu et al., 2023), QLoRA (Dettmers et al., 2024). Before training, we calibrate saliency and acquire a 2-bit weight spiking model with the EfficientQAT (Chen et al., 2024) setting. We follow the same settings of QLoRA and IR-LoRA in Alpaca finetuning.

Table 11: Performance comparison in QLoRA Settings.

| Method | Data | #Bit | Hums. | STEM | Social | Other | Avg. |
|---|---|---|---|---|---|---|---|
| LLAMA-7B | - | 16 | 33.3 | 29.8 | 37.8 | 38.0 | 34.6 |
| NormalFloat | - | 2 | 24.2 | 28.9 | 31.1 | 25.0 | 26.9 |
| QLoRA (w/ GPTQ) | Alpaca | 2 | 23.4 | 26.2 | 26.4 | 28.4 | 25.8 |
| QLoRA | Alpaca | 2 | 24.0 | 27.0 | 27.5 | 26.7 | 26.2 |
| QA-LoRA | Alpaca | 2 | 27.3 | 26.1 | 26.1 | 30.3 | 27.5 |
| IR-QLoRA | Alpaca | 2 | 26.0 | 27.8 | 30.2 | 28.3 | 27.8 |
| SpikeLLM-QLoRA$_{\tau=2}$ | Alpaca | 2 | 31.5 | 27.5 | 29.5 | 32.0 | 30.3 |

### A.5.3 COMPARISON IN EFFICIENT QAT SETTINGS

For low-bit weight-only quantizations, we train and evaluate SpikeLLM following EfficientQAT (Chen et al., 2024) (including training data and evaluation metrics, lm-eval-v0.4.2). Both the EfficientQAT and SpikeLLM are trained in 2048 length in two stages with the same condition. We compare the 2-bit performance as follows.

Table 12: Performance comparison in Efficient QAT pipeline.

| Method | #Bit | WinoGrande | HellaSwag | ArcC | ArcE | PiQA | Avg. |
|---|---|---|---|---|---|---|---|
| LLAMA-2-7B | - | 69.22 | 57.18 | 43.17 | 76.09 | 78.24 | 64.78 |
| GPTQ | 2 | 55.17 | 32.59 | 21.25 | 40.45 | 58.32 | 41.56 |
| OmniQ | 2 | 55.88 | 40.28 | 23.46 | 50.13 | 65.13 | 46.98 |
| AutoRound | 2 | 61.01 | 40.28 | 32.25 | 65.99 | 72.96 | 54.50 |
| AQLM$_{2\times8}$ | 2 | 65.27 | 49.96 | 32.85 | 66.92 | 73.07 | 57.61 |
| EfficientQAT | 2 | 65.19 | 49.53 | 35.15 | 69.74 | 73.67 | 58.65 |
| SpikeLLM$_{\tau=2}$ | 2 | 65.35 | 50.90 | 36.01 | 70.54 | 74.54 | 59.47 |

