# OpenReview forum: "SpikeLLM: Scaling up Spiking Neural Network to Large Language Models via Saliency-based Spiking"
_ICLR.cc/2025/Conference — ICLR 2025 Poster_

### Official Review · Reviewer_2u4J · 2024-10-18

**Soundness:** 3
**Presentation:** 3
**Contribution:** 3
**Rating:** 6
**Confidence:** 4

**Summary:**

This paper proposes the first large-scale spiking large language model that can scale the spiking neuron dynamics to handle models with 7 billion to 70 billion parameters. It has the potential to address the energy and computational resource limitations typically associated with running large language models. The main contributions proposed by the authors include:

1. Generalized Integral-Fire (GIF) neurons: This model compresses the spiking length by encoding the pulse train more efficiently. These neurons reduce the spiking length from T bits to T/L*log_2L bits. This novel approach improves coding efficiency while retaining the necessary computational power.

3. Optimal Brain Spiking (OBS) framework:a saliency-based spiking mechanism that dynamically allocates spiking steps based on the importance of channels within the network. Channels that are considered more important receive more spike steps to capture information, while less important channels receive fewer steps. This saliency-based approach enables more efficient quantization and information encoding, further improving energy efficiency.

4. Compatibility with modern hardware: SpikeLLM's GIF neurons and OBS framework can be deployed using both general matrix multiplication (GEMM) of traditional GPUs and event-driven neuromorphic chip architectures. This increases the feasibility of this method for deployment on neuromorphic hardware.

5. Experimental results: Through various benchmarks, SpikeLLM has significantly improved both performance and energy efficiency over traditional quantization methods used in LLM, such as OmniQuant and GPTQ pipelines. For example, it reduces the perplexity on WikiText2 by 24.85% and improves the common sense reasoning accuracy by 2.01% compared to the LLAMA2-7B model with standard quantization.

**Strengths:**

1. This paper attempts to solve the energy efficiency problem of LLM through SNN. This is a widely concerned issue. The author proposed several SNN based methods to solve the problems faced in traditional model quantization. The research problem is important and attractive, and the methods and experiments are of practical significance.

2.The author proposed three core methods: GLIF, SALIENCY-AWARE SPIKING STEPS (SASS), and OPTIMAL BRAIN SPIKING (OBP). The logic is clear and easy to follow:
-GLIF tries to encode more information in a single spike time step by merging L spiking steps
-SASS distinguishes the importance of different channels and assigns different quantization levels to different channels to balance the computational load and performance
-OBS determines the importance of any weights and activations through gradient methods.

3.According to the author's experiments, the proposed pipeline works well and outperforms the quantized ANN of the same level in terms of energy efficiency and performance. The authors conducted extensive testing on both the diversity of experimental datasets and model sizes.

**Weaknesses:**

1.Please explain/correct Eq3, it seems confusing to me.
2.In eq5, please fix the summation over t
3.Fig4(a) missing baseline label

My main concerns are about the computational complexity of the proposed method. The authors did not elaborate in detail in the article on the computational cost of the proposed GLIF, SASS, and OPTIMAL BRAIN SPIKING. Specifically:

4. For GLIF, In eq5, merging IF neurons of multiple steps into one step does reduce the requirement for SNN running time, but increases the computational cost of a single neuron; on the other hand, the implementation of neuromorphic hardware with multiple quantization levels is also unclear. Therefore, it is not sufficient to measure only auto-regressive steps here. Please provide a more detailed demonstration of the advantages of the proposed method.

5. For SASS and OBP,  Does the saliency calculation need to be performed both during training and inference? What is the additional computational cost? How can KV-caches be combined with the proposed method to further reduce the amount of computation? According to my understanding, eq10 only needs to be calculated once, but eq7 and 9 must be performed during inference. (point out if I am wrong) Please  elaborate on the additional computational cost.

Experiment result:

6. In table4, Most of the experimental results for 70b models are missing. The authors suggest that GQA makes training unstable. It would be good to have a more detailed analysis showing the key reasons and potential research directions here. In other words: how scalable is the approach proposed by the authors? What specific issues need to be addressed for stable scaling to 70b or larger models?

7. In table3, Although the proposed method is better than the traditional quantization method, the performance gap with the original model is still more than 10%, which limits the practicality of the method. The author proposed in Limitations: Continuous pretraining would boost the performance. What are the limits of both ann quantization and spiking llm boost? What is the performance difference with sufficient training time?

**Questions:**

See weakness. Listed below:
1.Please explain/correct Eq3, it seems confusing to me.
2.In eq5, please fix the summation over t
3.Fig4(a) missing baseline label
4.Please provide a more detailed demonstration of the advantages of the proposed method.
5.For SASS and OBP,  Does the saliency calculation need to be performed both during training and inference? etc
6.how scalable is the approach proposed by the authors? etc
7.What are the limits of both ann quantization and spiking llm boost? etc

---

> ### Author Response · Authors · 2024-11-23
>
> >1.Please explain/correct Eq3, it seems confusing to me.
>
> Thank you for your valuable advice regarding the writing. Eq. 3 is right and we have revised to make it more comprehensible. The previous SNN implementation was primarily designed for ReLU activations, which are non-negative and commonly used in CNN models. In this context, the spike output can be represented as:
> $ \mathbf{x}^{(\ell-1)}(t)= \mathbf{s}^{(\ell)}(t)V_{th} $.
> However, for Transformer models, where most linear layers handle activations with numerous negative values, it is challenging to directly simulate these with spiking IF neurons. Therefore, we have made the following adjustments:
>
> (i) For non-negative activations, we subtract their possible minimum value before spike encoding:
> $ \mathbf{x}^+ = \mathbf{x} - \mathbf{x_{min}} $,
> where $ \mathbf{x_{min}} $ is determined during training by statistically computing the minimum value for a specific token. This value is then fixed during inference as a quantization zero point.
>
> (ii) Only the positive part $ \mathbf{x}^+ $ is encoded into spikes, which are then used for matrix computations in the linear layer:
> $ \mathbf{y}^+ = \mathbf{x}^+W $.
>
> (iii) In the output stage, since $ \mathbf{x_{min}} $ does not participate in the linear layer calculations, it is added back after the spike-driven linear layer computation to maintain numerical equivalence:
> $ \mathbf{y} = \mathbf{x}^+W + \mathbf{x_{min}}W $,
> where $ \mathbf{x_{min}}W $ is a fixed value during inference and can be precomputed.
>
> We hope these explanation address the issue effectively.
>
> >2.In eq5, please fix the summation over t
>
> Thank you for the reminder. We have changed $t$ to $t'$, where $t' = \lfloor t / L \rfloor$. To describe Eq. 5 more clearly, we have added Fig. 7, which is located in Appendix A.4.1.
>
> >3.Fig4(a) missing baseline label
>
> Thank you. We have updated the baseline label in Fig. 4(a) to represent the percentage of saliency channels.
>
> >4. For GLIF, In eq5, merging IF neurons of multiple steps into one step does reduce the requirement for SNN running time, but increases the computational cost of a single neuron; on the other hand, the implementation of neuromorphic hardware with multiple quantization levels is also unclear. Therefore, it is not sufficient to measure only auto-regressive steps here. Please provide a more detailed demonstration of the advantages of the proposed method.
>
> We explain the advantages of GIF neurons from both the training and inference perspectives. To provide a clearer explanation, we have added Appendix 4.2 to describe the inference process of GIF neurons and Appendix 4.3 to compare GIF neurons with directly trained SNNs.
>
> **Traning** For large-scale SNN-based language models, training challenges have been a significant bottleneck, primarily due to the following two factors:
>
> (i) The high memory and time cost of training large-scale SNNs:
> Training an SNN for $T$ steps using direct training requires $T$ times the activation memory and training time of an equivalent ANN. By aggregating $L$ steps using GIF neurons, the activation memory consumption during training is reduced by a factor of $L$, and the training speed is accelerated by $L$ times. Consequently, $L$ times more training data can be processed in the same amount of time. Given the data-hungry nature of large models, GIF neurons make it easier to train large-scale SNNs effectively.
>
> (ii) Error accumulation in directly trained SNNs using BPTT:
> Directly trained SNNs rely on backpropagation through time (BPTT), where quantization errors at each time step accumulate over long temporal durations, often resulting in training failure. Fig. 9 compares the training dynamics of conventional spiking neural networks (SNNs) and GIF neurons:
>
> - Fig. 9(a): Traditional BPTT for IF (or LIF) neurons suffers from significant quantization errors due to binary quantization at each time step. These errors accumulate over time, making it difficult to train SNNs.
> - Fig. 9(b): GIF neurons adopt quantization-aware training. By aggregating $L$ time steps, GIF neurons perform multi-bit quantization, which eliminates inter-step error accumulation. This multi-bit quantization improves gradient estimation through the straight-through estimator (STE), enabling more accurate and stable training.
>
> **Inference** GIF neurons support both multi-bit and 1-bit inference modes:
>
> - 1-bit inference: Appendix 4.2 demonstrates the equivalence between IF neurons and GIF neurons under this mode.
> - Multi-bit inference: GIF neurons use a bit width of $\log L$ and can represent the information of $L$ IF neuron time steps in a single step. This reduces the total bit budget by a factor of $(L / \log L > 1)$ (e.g., when $L = 4$, the total bit budget is reduced by 2x; when $L = 16$, it is reduced by 4x).

---

> ### Author Response · Authors · 2024-11-23
>
> >For SASS and OBP, Does the saliency calculation need to be performed both during training and inference? What is the additional computational cost? How can KV-caches be combined with the proposed method to further reduce the amount of computation? According to my understanding, eq10 only needs to be calculated once, but eq7 and 9 must be performed during inference. (point out if I am wrong) Please elaborate on the additional computational cost.
>
> There are only additional computational cost in training for salieney calculation, for both SASS and OBP (or OBSpiking in paper). And there is no additional cost to calculate saliency in inference. After training, we can store these funded salient channels as a per-channel mask for each layer. And also, the mask is lightweight to store (for example, if the size of weights is 4096x4096 in Llama-7B, the saliency mask size is 4096x1). So, feel free to apply saliency detection.
>
> During training, we use Eq. 7 to describe the distinction between salient and non-salient channels conceptually, without performing any actual computations. In the OBSpiking framework, Eq. 9 and Eq. 10 are used to calculate the saliency of activations and weights, respectively.
>
> In our experiments, saliency is calculated using only 128 samples, and this computation is performed just once. Therefore, the additional computational cost during training is negligible. This is because saliency requires only a single inference pass with 128 calibration samples, while model parameter training involves multiple optimization iterations.
>
> If you have any further questions, please feel free to let us know.
>
> >How can KV-caches be combined with the proposed method to further reduce the amount of computation?
>
> SpikeLLM has implemented spiking operations for KV-cache to enable KV-cache compression, while also reducing operations in self-attention. Please refer to Figure 2 (Left) and lines 296–300.
>
> In detail, GIF neurons are inserted before the KV-Cache. The KV-Caches can be simulated by spikes the same as activations in linear layers. And then, we use the proposed OBSpiking to detect salient channels in training, and conduct saliency-based spiking in inference. In our experiments, as shown in Figure 4 (b), we have ablation studies to prove the effectiveness of saliency-based spiking in KV-Caches, and also achieve better performance.
>
> >In table4, Most of the experimental results for 70b models are missing. The authors suggest that GQA makes training unstable. It would be good to have a more detailed analysis showing the key reasons and potential research directions here. In other words: how scalable is the approach proposed by the authors? What specific issues need to be addressed for stable scaling to 70b or larger models?
>
> The instability of GQA optimization arises from two primary reasons:
>
> (i) GQA itself is a compression technique for KV-cache. Applying additional compression methods such as quantization-aware training or spiking operations can introduce significant quantization errors.
>
> (ii) In this paper, we adopt a layer-by-layer training approach, which is a resource-efficient method tailored for constrained scenarios. However, this approach also has inherent stability issues. Specifically, we follow the setup of OmniQuant, performing layer-wise fine-tuning for each transformer block of the ANN. The advantage of this method is that it requires training only one transformer layer at a time, significantly reducing storage and computational time. However, any errors introduced during the training of earlier layers can propagate and accumulate as quantization errors across subsequent layers, leading to training instability.
>
> For the 70B model, the primary cause of training instability is the increased quantization error from GQA's compression, which is further amplified by the error accumulation inherent in the layer-by-layer training pipeline.
>
> For future training of large-scale SNN-based LLMs, this paper provides the following conclusions:
>
> - End-to-end training is essential for larger SNN-LLMs. As the model scale increases, the required computational resources must also scale to ensure training stability for larger models.
> - On the other hand, this paper represents the first attempt to train SNN-LLMs exceeding 10B parameters. We demonstrate that using an efficient layer-by-layer training method, it is possible to scale SNNs to the 10B level.
> - Finally, the advantages of efficient training are also evident. For example, training a 7B sized SNN from Llama2-7B requires only 3.5 GPU hours.

---

> ### Author Response · Authors · 2024-11-24
>
> >In table3, Although the proposed method is better than the traditional quantization method, the performance gap with the original model is still more than 10%, which limits the practicality of the method. The author proposed in Limitations: Continuous pretraining would boost the performance. What are the limits of both ann quantization and spiking llm boost? What is the performance difference with sufficient training time?
>
> We have 3 best practices recently.
>
> (i) Using a more precise approach to eliminate outliers: Current state-of-the-art (SoTA) large model quantization methods employ matrix rotation to address outliers. However, as observed in Appendix 5, matrix rotation does not completely eliminate outliers in large models. To tackle the issue of massive outliers in QuaRot [1], we leverage SpikeLLM. We follow the same evaluations with QuaRot. More details and finding are shown in **Appendix A.5**. The results show that SpikeLLM surpasses SoTA quantization methods, achieving performance that closely approximates that of ANN-LLMs. For example, for the Llama-2-7B, SpikeLLM with the QuaRot backend is able to narrow the performance gap between ANNs and SNNs to 4\% (SpikeLLM-GPTQ).
>
> | **Method**         | **Saliency** | **#Bits** | **PIQA$_{\text{norm}}$** | **ARC-e$_{\text{norm}}$** | **ARC-c$_{\text{norm}}$** | **BoolQ** | **HellaSwag$_{\text{norm}}$** | **Winogrande** | **Avg.**  |
> |--------------------|--------------|-----------|--------------------------|---------------------------|---------------------------|-----------|------------------------------|----------------|-----------|
> | Llama-2-7b         | -            | FP16      | 79.11                    | 74.54                     | 46.42                     | 77.77     | 75.99                        | 69.06          | 70.48     |
> | QuaRot-RTN         | -            | W4A4      | 71.82                    | 59.89                     | 36.18                     | 67.37     | 63.88                        | 59.12          | 59.71     |
> | SpikeLLM-RTN       | 0.07         | W4A4      | 73.07                    | 61.99                     | 36.26                     | 68.81     | 64.15                        | 59.12          | **60.57** |
> | QuaRot-GPTQ        | -            | W4A4      | 75.95                    | 68.43                     | 39.76                     | 72.54     | 72.23                        | 64.72          | 65.61     |
> | SpikeLLM-GPTQ      | 0.08         | W4A4      | 76.77                    | 69.91                     | 42.06                     | 72.39     | 72.30                        | 65.43          | **66.48** |
> | Llama-2-13B        | -            | FP16      | 80.63                    | 77.48                     | 49.23                     | 80.73     | 79.37                        | 71.74          | 80.69     |
> | QuaRot-RTN         | -            | W4A4      | 74.86                    | 69.19                     | 41.98                     | 72.54     | 70.35                        | 64.72          | 65.61     |
> | SpikeLLM-RTN       | 0.05         | W4A4      | 75.35                    | 69.19                     | 43.00                     | 73.09     | 70.73                        | 66.46          | **66.30** |
> | QuaRot-GPTQ        | -            | W4A4      | 77.91                    | 72.18                     | 46.16                     | 78.41     | 75.55                        | 68.82          | 69.84     |
> | SpikeLLM-GPTQ      | 0.04         | W4A4      | 78.51                    | 71.89                     | 47.27                     | 79.02     | 75.77                        | 69.38          | **70.31** |
>
> ----------- to continue -------------

---

> ### Author Response · Authors · 2024-11-24
>
> (ii) Another best practice is training SpikeLLM with more data.
>
> We increase training data from 0.2M tokens to 8M, and using the EfficientQAT[2] framework for spiking weight quantization. We follow the evaluation protocol of EfficientQAT and report arc (not arc_norm) metrics as follows, which narrows performance gap to 5.38\%.
>
> | **Method**              | **#Bit** | **WinoGrande** | **HellaSwag** | **ArcC** | **ArcE** | **PiQA** | **Avg.**  |
> |-------------------------|----------|----------------|---------------|----------|----------|----------|-----------------------|
> | Llama-2-7B                      | -        | 69.22          | 57.16         | 43.52    | 76.26    | 78.07    | 64.85                 |
> | GPTQ                    | 2        | 55.17          | 32.59         | 21.25    | 40.45    | 58.32    | 41.56                 |
> | OmniQ                   | 2        | 55.88          | 40.28         | 23.46    | 50.13    | 65.13    | 46.98                 |
> | AutoRound               | 2        | 61.01          | 40.28         | 32.25    | 65.99    | 72.96    | 54.50                 |
> | AQLM\(_{2x8}\)          | 2        | 65.27          | 49.96         | 32.85    | 66.92    | 73.07    | 57.61                 |
> | EfficientQAT        | 2        | 65.19          | 49.53         | 35.15    | 69.74    | 73.67    | 58.65                 |
> | **SpikeLLM$_{\texttt{T=2}}$** | **2**    | **65.35**      | **50.90**     | **36.01** | **70.54** | **74.54** | **59.47**            |
>
> (iii) We also try to fintune SpikeLLM with the QLoRA methods to enhance SpikeLLM in downstream tasks.
>
> Before training, we calibrate saliency and acquire a 2-bit weight spiking model with the EfficientQAT [5] setting. We follow the same setting of QLoRA and IR-LoRA in Alpaca finetuning. SpikeLLM adding a QLoRA branch exceeds recent methods.
> | **Method** | **Data** | **#Bit** | **Hums.** | **STEM** | **Social** | **Other** | **Avg.** |
> |-----------------------------|----------|----------|-----------|----------|------------|-----------|----------|
> | LLaMA-7B | - | 16 | 33.3 | 29.8 | 37.8 | 38.0 | 34.6 |
> | NormalFloat                | -        | 2        | 24.2      | 28.9     | 31.1       | 25.0      | 26.9     |
> | QLoRA (w/ GPTQ)            | Alpaca   | 2        | 23.4      | 26.2     | 26.4       | 28.4      | 25.8     |
> | QLoRA                      | Alpaca   | 2        | 24.0      | 27.0     | 27.5       | 26.7      | 26.2     |
> | QA-LoRA                    | Alpaca   | 2        | 27.3      | 26.1     | 26.1       | 30.3      | 27.5     |
> | IR-QLoRA                   | Alpaca   | 2        | 26.0      | **27.8**     | **30.2**       | 28.3      | 27.8     |
> | **SpikeLLM-QLoRA$_{\texttt{T=2}}$**   | Alpaca   | 2        | **31.5**      | 27.5     | 29.5       | **32.0**      | **30.3**     |
>
> These results have shown that SpikeLLM can narrow the performance gap to 4~5\% with more training data or refined training process, which demonstrated the practicality of SpikeLLM.
>
> **Reference**
>
> [1] Ashkboos S, Mohtashami A, Croci M L, et al. Quarot: Outlier-free 4-bit inference in rotated llms[J]. arXiv preprint arXiv:2404.00456, 2024.
>
> [2] Chen M, Shao W, Xu P, et al. Efficientqat: Efficient quantization-aware training for large language models[J]. arXiv preprint arXiv:2407.11062, 2024.

---

> > ### Comment · Reviewer_2u4J · 2024-11-24
> > **Raised the rating from 5 to 6.**
> >
> > Thanks to the author for the responses. These answers largely solved my problem. I have raised the rating from 5 to 6.

---

### Official Review · Reviewer_JYwS · 2024-10-29

**Soundness:** 3
**Presentation:** 3
**Contribution:** 3
**Rating:** 8
**Confidence:** 4

**Summary:**

The paper introduces a post-training quantization scheme for large language models (LLMs). The quantization function is computed by a spiking neural network. The contributions over prior works are a reduction of the requires number SNN simulation steps by grouping time-steps into low-bit integer operations, handling outliers by allowing certain channels to conduct multiple simulation steps, and detecting outliers in weights and activations offline. Results are compared to the OmniQuant baseline and show improved downstream task performance on a standard evaluation protocol.

**Strengths:**

The paper is technically sound, well written and clear in its presentation. Tables 3 and 4 meet the standards of evaluation of the LLM community. Figures 1 and 2 assist the reader in following the presented method. The extend of the material is on par with recent quantization works that have been published recently in top ML conferences (e.g. [1], [2]). The authors compare their method on the standard evaluation benchmarks for LLMs reporting both downstream task performance and language modeling performance of large scale models from LLAMA 7B to 70B.

ANN-SNN conversion as a research field is of relevance to the ICLR community. Prior methods were not evaluated at this large scale. It is expected that ANN-SNN conversion methods would face similar issues regarding outliers as classical LLMs do. The authors use the core component of SNNs, i.e. temporal processing, to counter outliers and to compute channels with potential outliers in higher precision through longer simulation time. It is worth pointing out that using higher precision for such channels has been prominently studied in the literature (e.g. [3]), and saliency-driven calibration has been studied for quantized LLM (e.g. in [4]). Also ANN-SNN conversion based on integrate-and-fire neurons that model increased precision by conducting multiple steps was presented in prior works (e.g. [5], [6]). The core contribution of this paper is therefore the combination of these concepts to demonstrate > 7B parameter LLM quantization by handling outliers with increased simulation time budgets for certain channels. The paper therefore allows the comparison of rate-coded spiking neural networks against classical quantization methods in the large scale domain of LLMs.

1. [SmoothQuant: Accurate and Efficient Post-Training Quantization for Large Language Models](https://arxiv.org/abs/2211.10438)
2. [OmniQuant: Omnidirectionally Calibrated Quantization for Large Language Models](https://arxiv.org/abs/2308.13137)
3. [LLM.int8(): 8-bit Matrix Multiplication for Transformers at Scale](https://arxiv.org/abs/2208.07339)
4. [SliM-LLM: Salience-Driven Mixed-Precision Quantization for Large Language Models](https://arxiv.org/abs/2405.14917)
5. [SpikeLM: Towards General Spike-Driven Language Modeling via Elastic Bi-Spiking Mechanisms](https://arxiv.org/abs/2406.03287)
6. [SpikeZIP-TF: Conversion is All You Need for Transformer-based SNN](https://arxiv.org/abs/2406.03470)

**Weaknesses:**

The fields of ANN-SNN conversion and LLM quantization are quickly moving. It is hence important to keep track of the recent developments in these fields. A weakness of this paper is that it is missing comparisons against some closely related recent papers.

[1. SpikeZIP-TF: Conversion is All You Need for Transformer-based SNN](https://arxiv.org/abs/2406.03470)
Related work that uses integrate-and-fire neurons to implement low-bit quantization in transformers. Arguably this work does not work with as large models as the present paper.

[2. SliM-LLM: Salience-Driven Mixed-Precision Quantization for Large Language Models](https://arxiv.org/abs/2405.14917)
Improves over OmniQuant with saliency based metrics, similar to the ones proposed here. The reviewer is missing a comparison in tables 3 and 4.

[3. QuaRot: Outlier-Free 4-Bit Inference in Rotated LLMs](https://arxiv.org/abs/2404.00456)
Shows that outliers can mostly be eliminated by mathematically equivalent parameterizations of LLMs (through appropriate weight matrix rotations). This work challenges some key claims made by the presented paper. E.g. line 370 claims "necessity to introduce SpikeLLM due to the inefficiency and incompatibility [...] of existing ANNs based quantisation strategies to spike neural networks". If outliers can be eliminated in LLMs, it seems likely that basic ANN-SNN conversion methods such as SpikeZIP would work in low precision even for larger LLMs.

**Questions:**

- Is it feasible to compute the Hessian for such large models? Are any approximations used to get the saliency from the Hessian?
- A related work paragraph on ANN-SNN conversion in the context of transformer based models would add value to the paper
- The reviewer suggests to reconsider the use of the term "auto-regressive" as a synonym for recurrent, recursive or sequential. Auto-regression is a particular processing paradigm for sequences that aim to regress themselves, and is usually not interchangeably used with terms like recurrent or recursive.
- The reviewer suggests to reconsider the claims on "necessity for introducing SpikeLLM" e.g. in lines 25, 110, 120, 370 in face of works like QuaRot that show that one get's away with simply rotating the weights.
- It seems like the character "\~" is out of place between 7B~70B in multiple places. Perhaps use `\sim` instead
- Please incorporate QuaRot and SliM-LLM in both the related work and the comparision

---

> ### Author Response · Authors · 2024-11-23
> **Comparisons against some closely related recent papers**
>
> >The fields of ANN-SNN conversion and LLM quantization are quickly moving. It is hence important to keep track of the recent developments in these fields. A weakness of this paper is that it is missing comparisons against some closely related recent papers.
>
> (i) Comparison with QuaRot
>
> Thanks for your suggestion. We have observed some interesting phenomena in QuaRot (**Appendix A.5**), and our experiments show SpikeLLM can further improve the SoTA QuaRot with the same training condition. This indicates spiking neuronal dynamics can improve quantization in more contextural literature.
>
> To integrate SpikeLLM with QuaRot, we first visualized QuaRot’s activations (see Fig. 10,11), leading to the following observations:
>
> - Although QuaRot effectively mitigates outliers in the input layers of transformer modules (key-proj, query-proj, value-proj in self-attention; up-proj, gate-proj in MLP), we identified a critical issue: massive outliers (up to 100,000× larger magnitudes) persist in the output layers of each module (out-proj in self-attention; down-proj in MLP), as shown in Fig. 10.
>
> - Additionally, a small subset of attention heads exhibits significant saliency and magnitude, which QuaRot’s Hadamard transformations cannot fully address. This limitation arises because each attention head uses distinct softmax attention weights, while Hadamard transformations operate only within individual attention heads (Fig. 11).
>
> Based on these insights, we introduce saliency-aware spiking steps and OBSpiking, targeting only the out-proj in self-attention and down-proj in MLP. Specifically, we don't revise the most input layers (key-proj, query-proj, value-proj in self-attention; up-proj, gate-proj in MLP) and only set the saliency-based spiking in the output layer group including out-proj in self-attention and down-proj in MLP. Based on the observation that salient channels are clustered in specific attention heads (Fig. 11), we set salient channels head-wise for out-proj layers.
>
> As shown in the following Table, integrating SpikeLLM into the QuaRot pipeline improves performance and more effectively addresses outliers in LLMs.
>
> | **Method**         | **Saliency** | **#Bits** | **PIQA$_{\text{norm}}$** | **ARC-e$_{\text{norm}}$** | **ARC-c$_{\text{norm}}$** | **BoolQ** | **HellaSwag$_{\text{norm}}$** | **Winogrande** | **Avg.**  |
> |--------------------|--------------|-----------|--------------------------|---------------------------|---------------------------|-----------|------------------------------|----------------|-----------|
> | Llama-2-7b         | -            | FP16      | 79.11                    | 74.54                     | 46.42                     | 77.77     | 75.99                        | 69.06          | 70.48     |
> | QuaRot-RTN         | -            | W4A4      | 71.82                    | 59.89                     | 36.18                     | 67.37     | 63.88                        | 59.12          | 59.71     |
> | SpikeLLM-RTN       | 0.07         | W4A4      | 73.07                    | 61.99                     | 36.26                     | 68.81     | 64.15                        | 59.12          | **60.57** |
> | QuaRot-GPTQ        | -            | W4A4      | 75.95                    | 68.43                     | 39.76                     | 72.54     | 72.23                        | 64.72          | 65.61     |
> | SpikeLLM-GPTQ      | 0.08         | W4A4      | 76.77                    | 69.91                     | 42.06                     | 72.39     | 72.30                        | 65.43          | **66.48** |
> | Llama-2-13B        | -            | FP16      | 80.63                    | 77.48                     | 49.23                     | 80.73     | 79.37                        | 71.74          | 80.69     |
> | QuaRot-RTN         | -            | W4A4      | 74.86                    | 69.19                     | 41.98                     | 72.54     | 70.35                        | 64.72          | 65.61     |
> | SpikeLLM-RTN       | 0.05         | W4A4      | 75.35                    | 69.19                     | 43.00                     | 73.09     | 70.73                        | 66.46          | **66.30** |
> | QuaRot-GPTQ        | -            | W4A4      | 77.91                    | 72.18                     | 46.16                     | 78.41     | 75.55                        | 68.82          | 69.84     |
> | SpikeLLM-GPTQ      | 0.04         | W4A4      | 78.51                    | 71.89                     | 47.27                     | 79.02     | 75.77                        | 69.38          | **70.31** |
>
> ----------- to continue -------------

---

> > ### Author Response · Authors · 2024-11-23
> >
> > **Reference**
> >
> > [1] Hassibi B, Stork D. Second order derivatives for network pruning: Optimal brain surgeon[J]. Advances in neural information processing systems, 1992, 5.
> >
> > [2] Qin H, Ma X, Zheng X, et al. Accurate lora-finetuning quantization of llms via information retention[J]. arXiv preprint arXiv:2402.05445, 2024.
> >
> > [3] Xu Y, Xie L, Gu X, et al. Qa-lora: Quantization-aware low-rank adaptation of large language models[J]. arXiv preprint arXiv:2309.14717, 2023.
> >
> > [4] Dettmers T, Pagnoni A, Holtzman A, et al. Qlora: Efficient finetuning of quantized llms[J]. Advances in Neural Information Processing Systems, 2024, 36.
> >
> > [5] Chen M, Shao W, Xu P, et al. Efficientqat: Efficient quantization-aware training for large language models[J]. arXiv preprint arXiv:2407.11062, 2024.

---

> > > ### Comment · Reviewer_JYwS · 2024-11-27
> > > **Raising the score**
> > >
> > > The authors' additional evaluation provided in the rebuttal is highly appreciated, and addresses all raised concerns. I raised the score to 8.
> > >
> > > Do the authors intend to add the data provided during the rebuttal to the main body of the paper? Unrelated to this questions, the observation of persistent outliers in QuaRot is quite interesting and might be of interest to a wider audience than what page 19/20 will eventually reach.

---

> > > > ### Author Response · Authors · 2024-11-28
> > > > **Correction of the Down-Proj Visualization in the Added Experiment, Fig. 11**
> > > >
> > > > Dear Reviewer,
> > > >
> > > > Yesterday, we discovered a visualization error (in QuaRot, Fig. 11) regarding the MLP's down-projection, which doesn't influence the overall conclusion.
> > > > Our corrections are as follows:
> > > >
> > > > - In QuaRot, the most significant outliers appear in the self-attention mechanism's value (cache) and out-projection layers. Salient channels are clustered in certain attention heads. (The down-projection is not as salient after correction.)
> > > > - Therefore, we add saliency-based spiking only in the value and out-proj layers, according to salient attention heads. We have rerun the experiments (Table 10): removing the saliency-based spiking from the down-proj layer has negligible impact on the results of SpikeLLM, and it reduces the total number of spikes.
> > > >
> > > > We have updated the visualization of the value cache and out-proj layers in Fig. 11. The corrected results have been uploaded and do not affect the overall results compared with QuaRot.
> > > >
> > > > Best Regards,
> > > >
> > > > The Authors

---

> ### Author Response · Authors · 2024-11-23
> **Comparisons against some closely related recent papers**
>
> (ii) Comparison with SliM-LLM
>
> (a) Saliency Detection Method
>
> SliM-LLM primarily relies on traditional OBS (Optimal Brain Surgeon), which allocates saliency to weights based on the Hessian matrix.
> Our key discovery is that for a well-optimized LLM, the first-order saliency of activations is non-zero (unlike the first-order saliency of weights, which is zero as per OBS [1]). This observation allows us to bypass the computation of the Hessian matrix by directly using first-order saliency for activations.
>
> Thus, we propose OBSpiking, an extension of traditional OBS that supports saliency detection for activations. The final saliency calculation depends on the context:
>
> - For activation quantization: We use the first-order saliency of activations, avoiding the Hessian matrix computation.
> - For weight quantization: We adopt traditional OBS [1], similar to SliM-LLM.
>
> (b) Acceleration Performance
>
> - SpikeLLM focuses on structured spiking allocation. Using spiking dynamics (Eqs. 1, 3, 5), mixed-precision quantization can be simplified into single-precision quantization. As shown in Fig. 1, spiking neurons transform mixed-precision quantization (Fig. 1(c)) into single-precision quantization (Fig. 1(d)), which can be directly accelerated using the GEMM kernel. Additionally, weight quantization also reduces memory requirements.
>
> - SliM-LLM, on the other hand, employs unstructured weight-only mix-precision quantization. During computation, it requires reconversion to FP16 for processing. Due to its unstructured approach, quantized computation is not feasible, limiting its benefits to memory savings.
>
> (c) Matching SoTA Weight-Only Quantization Performance
>
> To achieve state-of-the-art performance in the weight-only quantization setting, we improve SpikeLLM’s performance using the EfficientQAT backend. Both SpikeLLM and EfficientQAT are trained on sequences of length 2048. We follow the evaluation protocol of EfficientQAT and report arc (not arc_norm) metrics as follows.
>
> | **Method**              | **#Bit** | **WinoGrande** | **HellaSwag** | **ArcC** | **ArcE** | **PiQA** | **Avg.**  |
> |-------------------------|----------|----------------|---------------|----------|----------|----------|-----------------------|
> | Llama-2-7B                      | -        | 69.22          | 57.16         | 43.52    | 76.26    | 78.07    | 64.85                 |
> | GPTQ                    | 2        | 55.17          | 32.59         | 21.25    | 40.45    | 58.32    | 41.56                 |
> | OmniQ                   | 2        | 55.88          | 40.28         | 23.46    | 50.13    | 65.13    | 46.98                 |
> | AutoRound               | 2        | 61.01          | 40.28         | 32.25    | 65.99    | 72.96    | 54.50                 |
> | AQLM\(_{2x8}\)          | 2        | 65.27          | 49.96         | 32.85    | 66.92    | 73.07    | 57.61                 |
> | EfficientQAT        | 2        | 65.19          | 49.53         | 35.15    | 69.74    | 73.67    | 58.65                 |
> | **SpikeLLM$_{\texttt{T=2}}$** | **2**    | **65.35**      | **50.90**     | **36.01** | **70.54** | **74.54** | **59.47**            |
>
> Since the current best results and SliM-LLM use different test prompts, we compared the perplexity of Llama models on WikiText-2.
>
> | Model                 | Inference Type      | Precision       | #Bit | Llama-1-7B | Llama-2-7B |
> |-----------------------|---------------------|-----------------|------|------------|------------|
> | SliM-LLM+            | Unstructured        | Mix-Precision   | 2    | 9.68       | 10.87      |
> | SpikeLLM (EfficientQAT) | Structured          | Single-Precision | 2    | **7.34**       | **7.18**       |
>
> ----------- to continue -------------

---

> ### Author Response · Authors · 2024-11-23
> **Comparisons against some closely related recent papers and other questions**
>
> (iii) Comparison with SpikeZIP-FT
>
> The key difference between SpikeLLM and SpikeZIP-FT lies in the proposed OBSpiking framework, which enables SpikeLLM to achieve two capabilities that SpikeZIP-FT lacks:
>
> **Surpassing the SoTA QuaRot Quantization Method**
>
> - SpikeZIP-FT's training process involves two stages: first training the corresponding quantized artificial neural network (QANN) and then converting the quantized ANN into a spiking-driven inference form. However, this process inherently limits SpikeZIP-FT's performance to that of the corresponding quantized model, preventing it from exceeding the quantized model's performance.
>
> - SpikeLLM introduces a saliency-based spiking mechanism that enhances significant channels, thereby improving the quantized model's performance. Using this approach, SpikeLLM can exceed the performance of current state-of-the-art (SoTA) quantized models during QuaRot training—a feat that SpikeZIP-FT cannot achieve.
>
> **Addressing the ANN-SNN Conversion Outlier Problem**
>
> SpikeZIP-FT faces challenges with massive outliers during the ANN-to-SNN conversion process. Early LLM-QAT methods, despite leveraging large training datasets for end-to-end training, still suffered from significant quantization errors. Ignoring outliers can lead to high training resource consumption and hinder spiking large model training.
>
> SpikeLLM's OBSpiking framework effectively mitigates this issue and can be applied across various training methods, including PTQ, QAT, and QLoRA. We have extended SpikeLLM to these methods, achieving SoTA results:
>
> - SpikeLLM + QLoRA.
> This line of work focuses on adding a low-rank branch to finetune the low-bit quantized model. State-of-the-art methods including IR-QLoRA [2], QA-LoRA [3], QLoRA [4]. Before training, we calibrate saliency and acquire a 2-bit weight spiking model with the EfficientQAT [5] setting. We follow the same setting of QLoRA and IR-LoRA in Alpaca finetuning. SpikeLLM adding a QLoRA branch exceeds recent methods.
> | **Method** | **Data** | **#Bit** | **Hums.** | **STEM** | **Social** | **Other** | **Avg.** |
> |-----------------------------|----------|----------|-----------|----------|------------|-----------|----------|
> | LLaMA-7B | - | 16 | 33.3 | 29.8 | 37.8 | 38.0 | 34.6 |
> | NormalFloat                | -        | 2        | 24.2      | 28.9     | 31.1       | 25.0      | 26.9     |
> | QLoRA (w/ GPTQ)            | Alpaca   | 2        | 23.4      | 26.2     | 26.4       | 28.4      | 25.8     |
> | QLoRA                      | Alpaca   | 2        | 24.0      | 27.0     | 27.5       | 26.7      | 26.2     |
> | QA-LoRA                    | Alpaca   | 2        | 27.3      | 26.1     | 26.1       | 30.3      | 27.5     |
> | IR-QLoRA                   | Alpaca   | 2        | 26.0      | **27.8**     | **30.2**       | 28.3      | 27.8     |
> | **SpikeLLM-QLoRA$_{\texttt{T=2}}$**   | Alpaca   | 2        | **31.5**      | 27.5     | 29.5       | **32.0**      | **30.3**     |
> - SpikeLLM + EfficientQAT, please refer to (ii).
> - SpikeLLM + QuaRot (PTQ), please refer to (i).
>
> >Is it feasible to compute the Hessian for such large models? Are any approximations used to get the saliency from the Hessian?
>
> (i) Under the weight quantization setting, we independently compute the Hessian matrix for each linear layer based on the $L_2$ error. Thus, the calculation of the Hessian can be simplified as $H=2XX^T$.
>
> (ii) We compute the Hessian only during the calibration phase, while the inference phase uses a per-channel mask. Therefore, the Hessian matrix needs to be calculated only once during training.
>
> > A related work paragraph on ANN-SNN conversion in the context of transformer based models would add value to the paper.
>
> Thank you for your suggestions on the writing. We will update the next version as soon as possible, including related works on ANN-to-SNN conversion. Previously, SNN-to-ANN conversion mainly relied on quantized ANNs. The key difference in our work lies in the saliency-based spiking steps, which can outperform traditional quantization methods.
>
> >The reviewer suggests to reconsider the use of the term "auto-regressive" as a synonym for recurrent, recursive or sequential. Auto-regression is a particular processing paradigm for sequences that aim to regress themselves, and is usually not interchangeably used with terms like recurrent or recursive.
>
> Thank you very much for pointing this out. We hadn't noticed it before, and we will address the related expression issues in the next version. Additionally, the next version will incorporate the experimental results and analyses as mentioned above.

---

> ### Author Response · Authors · 2024-11-27
> **Thank you for reviewing**
>
> We will merge the experiments of QuaRot into the main paper and place the additional experiments mentioned above in the Appendix (because of the page limit). Due to time constraints, we will merge these results in the final version. Thank you for your time and constructive suggestions during the review process.

---

### Official Review · Reviewer_VrDb · 2024-10-31

**Soundness:** 3
**Presentation:** 3
**Contribution:** 4
**Rating:** 8
**Confidence:** 5

**Summary:**

This paper is the first to introduce SNNs into LLMs, an innovation that deserves high commendation and demonstrates the broad applicability of SNNs. Additionally, the authors combine quantization techniques with SNNs to propose GIF neurons and an OBS framework. Overall, I consider this paper to make a significant contribution.

**Strengths:**

1. This study is the first to apply SNNs to large-scale language models with up to 70 billion parameters, filling a gap in the field.
2.: The authors provide a good analysis of issues in quantized LLMs and propose a Spike-Driven Quantization method that effectively addresses these problems.
3. The proposed GIF neurons leverage the time steps of SNNs to mitigate outliers that may result from direct quantization.
4. The method is thoroughly trained and tested on LLAMA-2-7B and LLAMA-2-70B models, validating its effectiveness. To my knowledge, this is the first time that SNNs have been tested on a model as large as LLAMA-2-70B.

**Weaknesses:**

1\I strongly suggest that the author takes the time to offer a comprehensive and meticulous description of the precise experimental results. It is essential to highlight the pivotal discoveries and thoroughly explore their potential ramifications within the broader context of the study.
2\It would greatly enhance the transparency and accessibility of the research if the author decides to open source the code. This action would not only foster a collaborative environment but also enable other researchers to replicate and extend the findings, ultimately advancing the field.
3\To fully ascertain the efficacy and superiority of the proposed method, it is imperative that the author conducts a more extensive series of experiments. These experiments should rigorously compare the performance of the current approach with various artificial neural network (ANN) alternatives, providing a comprehensive analysis of their respective strengths and weaknesses.

**Questions:**

1. Lack of Logical Structure in Related Work: The related work section primarily lists existing studies without a systematic analysis of the development and challenges of SNNs and quantization methods.
2. Unclear Formula Description: The explanation of Equation 5 (lines 239-244) is not intuitive, making it difficult for readers to understand the specific meaning of the GLIF model.
3. Insufficient Visualization: There is a lack of intuitive graphical illustrations to demonstrate how the GLIF model divides long binary sequences into smaller segments over multiple time steps.

---

> ### Author Response · Authors · 2024-11-22
>
> >It would greatly enhance the transparency and accessibility of the research if the author decides to open source the code. This action would not only foster a collaborative environment but also enable other researchers to replicate and extend the findings, ultimately advancing the field. (W2)
>
> Thank you for your interest in this paper. We will open source the code on GitHub.
>
> >To fully ascertain the efficacy and superiority of the proposed method, it is imperative that the author conducts a more extensive series of experiments. These experiments should rigorously compare the performance of the current approach with various artificial neural network (ANN) alternatives, providing a comprehensive analysis of their respective strengths and weaknesses. (W3)
>
> Traditional ANN-LLM compression methods include model sparsity (such as prunning), low-bit quantization, low-rank decomposition, and distillation. Because distillation can be viewed as a training method, we mainly focus on coomparisons with recent LLM sparsity, SoTA LLM quantization, and low-rank decomposition methods.
>
> (i) Comparison with SoTA LLM quantization.
>
> As we addressed in the paper, SpikeLLM is general enough to cooperate with different quantization methods, which can be also viewed as improving traditional with spiking neuronal dynamics. In the following, we are delighted to show SpikeLLM is able to improve the recent SoTA QuaRot [1] with the same quantization technology, training and evaluation conditions (more analysis in Appendix A.5).
>
> | **Method**         | **Saliency** | **#Bits** | **PIQA$_{\text{norm}}$** | **ARC-e$_{\text{norm}}$** | **ARC-c$_{\text{norm}}$** | **BoolQ** | **HellaSwag$_{\text{norm}}$** | **Winogrande** | **Avg.**  |
> |--------------------|--------------|-----------|--------------------------|---------------------------|---------------------------|-----------|------------------------------|----------------|-----------|
> | Llama-2-7b         | -            | FP16      | 79.11                    | 74.54                     | 46.42                     | 77.77     | 75.99                        | 69.06          | 70.48     |
> | QuaRot-RTN         | -            | W4A4      | 71.82                    | 59.89                     | 36.18                     | 67.37     | 63.88                        | 59.12          | 59.71     |
> | SpikeLLM-RTN       | 0.07         | W4A4      | 73.07                    | 61.99                     | 36.26                     | 68.81     | 64.15                        | 59.12          | **60.57** |
> | QuaRot-GPTQ        | -            | W4A4      | 75.95                    | 68.43                     | 39.76                     | 72.54     | 72.23                        | 64.72          | 65.61     |
> | SpikeLLM-GPTQ      | 0.08         | W4A4      | 76.77                    | 69.91                     | 42.06                     | 72.39     | 72.30                        | 65.43          | **66.48** |
> | Llama-2-13B        | -            | FP16      | 80.63                    | 77.48                     | 49.23                     | 80.73     | 79.37                        | 71.74          | 80.69     |
> | QuaRot-RTN         | -            | W4A4      | 74.86                    | 69.19                     | 41.98                     | 72.54     | 70.35                        | 64.72          | 65.61     |
> | SpikeLLM-RTN       | 0.05         | W4A4      | 75.35                    | 69.19                     | 43.00                     | 73.09     | 70.73                        | 66.46          | **66.30** |
> | QuaRot-GPTQ        | -            | W4A4      | 77.91                    | 72.18                     | 46.16                     | 78.41     | 75.55                        | 68.82          | 69.84     |
> | SpikeLLM-GPTQ      | 0.04         | W4A4      | 78.51                    | 71.89                     | 47.27                     | 79.02     | 75.77                        | 69.38          | **70.31** |
>
> (ii) Comparison with LLM sparsity and LLM low-rank decomposition.
>
> The LLM sparsity mainly focuses on unstructured, semi-structured or structured sparse weights or activations. Because weights are easier to compress than activations. We mainly compare with the most popular weight pruning methods include SparseGPT [2], Wanda [3], SliceGPT [4], et al.
>
> Another line of work to compress weights is LLM low-rank decomposition, including SVD-LLM [5] and ASVD [6]. In general, direct model sparsity achieves higher performance than SVD methods in recent works.
>
> ----------- to continue -------------

---

> > ### Comment · Reviewer_VrDb · 2024-11-23
> >
> > Thank you for addressing my concerns. I will  increase my score and look forward to your open-source code

---

> ### Author Response · Authors · 2024-11-22
>
> To match SoTA performance, we train the weight quantized SpikeLLM with the QuaRot [1] backend. Because different evaluation metrics (and evaluation prompts) are employed in these methods,we only compare there preplexity on the wikitext2 dataset. All these results are based on the Llama-2-7B model.
>
> ||Method     | Compression Type | Compression Ratio | PPL  |
> |---|---------|------------------|-------------------|------|
> | SparseGPT  | Pruning          | Semi-structured   | x2 (2:4)     | 8.69 |
> | SliceGPT   | Pruning          | Structured        | x1.4          | 8.12 |
> | SVD-LLM    | SVD              | Structured        | x1.25         | 10.10 |
> | ASVD       | SVD              | Structured        | x1.25         | 8.50 |
> | SpikeLLM   | SNN/Quant.       | Structured        | x4 (4W4A4KV)  | 6.27 |
>
>
> (iii) Comparison with QLoRA [7] methods for downstream finetuning:
> This line of work focuses on adding a low-rank branch to finetune the low-bit quantized model. State-of-the-art methods including IR-QLoRA [8], QA-LoRA [9], QLoRA [7]. Before training, we calibrate saliency and acquire a 2-bit weight spiking model with the EfficientQAT [5] setting. We follow the same setting of QLoRA and IR-LoRA in Alpaca finetuning. SpikeLLM adding a QLoRA branch exceeds recent methods.
>
> | **Method** | **Data** | **#Bit** | **Hums.** | **STEM** | **Social** | **Other** | **Avg.** |
> |-----------------------------|----------|----------|-----------|----------|------------|-----------|----------|
> | LLaMA-7B | - | 16 | 33.3 | 29.8 | 37.8 | 38.0 | 34.6 |
> | NormalFloat                | -        | 2        | 24.2      | 28.9     | 31.1       | 25.0      | 26.9     |
> | QLoRA (w/ GPTQ)            | Alpaca   | 2        | 23.4      | 26.2     | 26.4       | 28.4      | 25.8     |
> | QLoRA                      | Alpaca   | 2        | 24.0      | 27.0     | 27.5       | 26.7      | 26.2     |
> | QA-LoRA                    | Alpaca   | 2        | 27.3      | 26.1     | 26.1       | 30.3      | 27.5     |
> | IR-QLoRA                   | Alpaca   | 2        | 26.0      | **27.8**     | **30.2**       | 28.3      | 27.8     |
> | **SpikeLLM-QLoRA$_{\texttt{T=2}}$**   | Alpaca   | 2        | **31.5**      | 27.5     | 29.5       | **32.0**      | **30.3**     |
>
> Based on these results, we draw the following conclusions:
>
> - Compared to traditional quantization, SpikeLLM enhances salient channels through spiking, effectively achieving mixed-precision quantization. Leveraging spiking neurons, it transforms mixed-precision quantization into single-precision quantization (Fig. 1). Saliency consistently improves the performance of traditional quantization.
> - Compared to SVD or sparsification methods, SpikeLLM enables significantly higher model compression ratios. In contrast, SVD and direct pruning achieve only limited compression, especially in structured model compression scenarios.
> - To demonstrate SpikeLLM's effectiveness in general downstream tasks, we combined SpikeLLM with the QLoRA method. It achieved the best results compared to similar approaches, proving SpikeLLM's versatility in fine-tuning.
>
> A limitation of this work is that SpikeLLM is not trained from scratch. Future work should explore methods for directly training spiking large language models while addressing the challenge of achieving greater biological plausibility.
>
> >Insufficient Visualization: There is a lack of intuitive graphical illustrations to demonstrate how the GLIF model divides long binary sequences into smaller segments over multiple time steps. (Q3)
>
>  As suggested, we illustrate details in **Appendix A.4**, with the spiking mechanism illustrated in Fig. 8 and training dynamics in Fig. 9.
>
> (i) Binery inference of GIF neurons (Fig. 8):
> - Neuronal Dynamics of IF Neurons (Fig. 8a): Given a real-valued input sequence, the membrane potential accumulates over time. When it exceeds the firing threshold, the IF neuron emits a spike, and the membrane potential is reset by subtracting the threshold value.
> - Neuronal Dynamics of Multi-Step GIF Neurons (Fig. 8b): Similar to IF neurons, GIF neurons accumulate membrane potential over time steps. However, unlike IF neurons, GIF neurons first integrate inputs over a fixed number of steps L. Based on Eq. 5, GIF neurons can fire k discrete levels of spikes, where $k \in [0, L]$.
> - Neuronal Dynamics of Binary GIF Neurons (Fig. 8c): The only difference between multi-bit and binary inference of GIF neurons lies in the firing threshold. GIF neurons in multi-bit inference have multiple thresholds corresponding to different firing levels, while binary inference uses a single threshold, similar to IF neurons. As a result, binary inference emits one bit of spike per time step, as a result, multi-level spikes effectively decoded as binary. Next, we demonstrate the equivalence of binary inferenced GIF neurons and IF neurons.
>
> ----------- to continue -------------

---

> ### Author Response · Authors · 2024-11-22
>
> (ii) Equivalence Conditions for Spiking Neurons:
> - Given the same real-valued input sequence with firing rate encoding, if the spike firing rates are identical, the neurons are considered equivalent.
> - For finite-step spikes, the membrane potential at the end of a given period must also be the same to ensure consistent firing rates in subsequent steps.
>
> Based on these conditions, since the binary GIF and IF neurons share the same inputs and firing threshold over a short time window, their firing rates and membrane potential values are identical, confirming their equivalence in binary inference.
>
> (iii) Comparison of Training Dynamics of conventional spiking neural networks (SNNs) and GIF neurons (Fig. 9):
> - Figure 9a: Traditional backpropagation through time (BPTT) for IF (or LIF) neurons suffers from significant quantization errors due to binary quantization at each time step. These errors accumulate over time, making SNNs difficult to train.
> - Figure 9b: GIF neurons use quantization-aware training. By aggregating L time steps, GIF neurons perform multi-bit quantization, which eliminates inter-step error accumulation. The multi-bit quantization improves gradient estimation (STE), leading to more accurate training.
>
> > Unclear Formula Description: The explanation of Equation 5 (lines 239-244) is not intuitive, making it difficult for readers to understand the specific meaning of the GLIF model. (Q2)
>
> Based on **Appendix A.4**, we hope the explanation of Equation 5 is shown clear in Fig. 8. If there are still questions, please do not hesitate to let us know.
>
> >Lack of Logical Structure in Related Work: The related work section primarily lists existing studies without a systematic analysis of the development and challenges of SNNs and quantization methods. (Q1)
>
> Thank you for your suggestions on our writing. We will incorporate content on the development of ANN and SNN encoding in the new version as soon as possible.
>
> (i) In the field of large models, quantization techniques are more closely related to the deployment of large models and have been extensively studied. However, there is still a lack of related research on SNNs in this area.
>
> (ii) Previous approaches to SNN-ANN conversion primarily relied on quantization-aware training, followed by simulation using SNNs, where the quantized model served as the performance ceiling for the SNN. In this work, by introducing the proposed spike steps based on saliency, we demonstrate that it is possible to surpass the performance of the corresponding quantization methods.
>
> **Reference**
>
> [1]  Ashkboos S, Mohtashami A, Croci M L, et al. Quarot: Outlier-free 4-bit inference in rotated llms[J]. arXiv preprint arXiv:2404.00456, 2024.
>
> [2] Frantar E, Alistarh D. Sparsegpt: Massive language models can be accurately pruned in one-shot[C]//International Conference on Machine Learning. PMLR, 2023: 10323-10337.
>
> [3] Sun M, Liu Z, Bair A, et al. A simple and effective pruning approach for large language models[J]. arXiv preprint arXiv:2306.11695, 2023.
>
> [4] Ashkboos S, Croci M L, Nascimento M G, et al. Slicegpt: Compress large language models by deleting rows and columns[J]. arXiv preprint arXiv:2401.15024, 2024.
>
> [5] Wang X, Zheng Y, Wan Z, et al. Svd-llm: Truncation-aware singular value decomposition for large language model compression[J]. arXiv preprint arXiv:2403.07378, 2024.
>
> [6] Yuan Z, Shang Y, Song Y, et al. Asvd: Activation-aware singular value decomposition for compressing large language models[J]. arXiv preprint arXiv:2312.05821, 2023.
>
> [7] Dettmers T, Pagnoni A, Holtzman A, et al. Qlora: Efficient finetuning of quantized llms[J]. Advances in Neural Information Processing Systems, 2024, 36.
>
> [8] Qin H, Ma X, Zheng X, et al. Accurate lora-finetuning quantization of llms via information retention[J]. arXiv preprint arXiv:2402.05445, 2024.
>
> [9] Xu Y, Xie L, Gu X, et al. Qa-lora: Quantization-aware low-rank adaptation of large language models[J]. arXiv preprint arXiv:2309.14717, 2023.

---

### Official Review · Reviewer_wFix · 2024-11-04

**Soundness:** 2
**Presentation:** 2
**Contribution:** 2
**Rating:** 6
**Confidence:** 4

**Summary:**

The authors propose SpikeLLM, an energy-efficient large language model designed to tackle the increasing energy demands of current LLMs. They highlight two approaches namely, Generalized Integrate-and-Fire (GIF) neurons  and Optimal Brain Spiking framework to compresses spike length. The experimental results show improvements compared to the baselines used in the paper.

**Strengths:**

The paper is well motivated and addresses a genuine problem of the increasing energy cost of LLMs. Experimental results show improved performance compared to the used baselines.

**Weaknesses:**

1) The temporal dynamics of the model is not explained in details. There is very limited results on the activity sparsity of the model, energy-efficiency comparison with other efficient LLMs (Table 5 only compares with LLAMA-1-7B)
2) The authors can use better baselines for comparison such as 1.58 bit LLM [1], IR-QLoRA[2].
3) There is no experimental results showing 1-bit inference on neuromorphic chips. Even simulation showing spiking activity and convergence to similar result (as the merged spike) will demonstrate that we can use the model for efficient on-chip inference. No code has been added as part of the experiment. It is hard to understand how one would run the network in a 1-bit setting. If 1-bit (activity) inference cannot be shown experimentally then we cannot actually call this method spiking.

References
[1] Ma, S., Wang, H., Ma, L., Wang, L., Wang, W., Huang, S., Dong, L., Wang, R., Xue, J. and Wei, F., 2024. The era of 1-bit llms: All large language models are in 1.58 bits. arXiv preprint arXiv:2402.17764.

[2] Qin, Haotong, Xudong Ma, Xingyu Zheng, Xiaoyang Li, Yang Zhang, Shouda Liu, Jie Luo, Xianglong Liu, and Michele Magno. "Accurate lora-finetuning quantization of llms via information retention." arXiv preprint arXiv:2402.05445 (2024).

**Questions:**

1) Since this approach assigns different values of T corresponding to saliency of particular channels, can you please explain the temporal dynamics of the model (preferably during the on chip 1-bit inference phase) with a schematic diagram?

2) In Table 6, SpikeLLM uses  ternary GIF neurons as weight quantizers. What is the quantization level of the activations?

3) The results shown in Table 3 for SpikeLLM variants, all have activation quantization levels >= 4. Can we really call the models spiking?

4) Can the authors please compare their results to current sota quantized LLMs.

Please see weaknesses as well.

---

> ### Author Response · Authors · 2024-11-22
>
> We appreciate your constructive comments on our paper and we hope the following results could clearly address your concerns. As you suggested, we conduct a wide comparison with SoTA quantization methods, putting SpikeLLM into a more general literature; and also append Figures to illustrate binary spiking dynamics.
>
> >There is no experimental results showing 1-bit inference on neuromorphic chips. Even simulation showing spiking activity and convergence to similar result (as the merged spike) will demonstrate that we can use the model for efficient on-chip inference. (W3)
>
> The proposed GIF neurons are equivalent in both binary inference and multi-bit inference. As suggested, we illustrate details in **Appendix A.4**, with the spiking mechanism illustrated in Fig. 8 and training dynamics in Fig. 9.
>
> (i) Binery inference of GIF neurons (Fig. 8):
> - Neuronal Dynamics of IF Neurons (Fig. 8a): Given a real-valued input sequence, the membrane potential accumulates over time. When it exceeds the firing threshold, the IF neuron emits a spike, and the membrane potential is reset by subtracting the threshold value.
> - Neuronal Dynamics of Multi-Step GIF Neurons (Fig. 8b): Similar to IF neurons, GIF neurons accumulate membrane potential over time steps. However, unlike IF neurons, GIF neurons first integrate inputs over a fixed number of steps L. Based on Eq. 5, GIF neurons can fire k discrete levels of spikes, where $k \in [0, L]$.
> - Neuronal Dynamics of Binary GIF Neurons (Fig. 8c): The only difference between multi-bit and binary inference of GIF neurons lies in the firing threshold. GIF neurons in multi-bit inference have multiple thresholds corresponding to different firing levels, while binary inference uses a single threshold, similar to IF neurons. As a result, binary inference emits one bit of spike per time step, as a result, multi-level spikes effectively decoded as binary. Next, we demonstrate the equivalence of binary inferenced GIF neurons and IF neurons.
>
> (ii) Equivalence Conditions for Spiking Neurons:
> - Given the same real-valued input sequence with firing rate encoding, if the spike firing rates are identical, the neurons are considered equivalent.
> - For finite-step spikes, the membrane potential at the end of a given period must also be the same to ensure consistent firing rates in subsequent steps.
>
> Based on these conditions, since the binary GIF and IF neurons share the same inputs and firing threshold over a short time window, their firing rates and membrane potential values are identical, confirming their equivalence in binary inference.
>
> (iii) Comparison of Training Dynamics of conventional spiking neural networks (SNNs) and GIF neurons (Fig. 9):
> - Figure 9a: Traditional backpropagation through time (BPTT) for IF (or LIF) neurons suffers from significant quantization errors due to binary quantization at each time step. These errors accumulate over time, making SNNs difficult to train.
> - Figure 9b: GIF neurons use quantization-aware training. By aggregating L time steps, GIF neurons perform multi-bit quantization, which eliminates inter-step error accumulation. The multi-bit quantization improves gradient estimation (STE), leading to more accurate training.
>
> >The results shown in Table 3 for SpikeLLM variants, all have activation quantization levels >= 4. Can we really call the models spiking? (Q3)
>
> We introduce spiking neuronal dynamics in two parts:
>
> (i) As shown in Fig. 8, the inference of SpikeLLM can be equivalently converted to binary spike inference.
>
> (ii) Regarding the relationship between multi-level spikes (within the same salient channels), they are encoded following the input equation, membrane potential accumulation, and output equation of SNNs.
>
> Moreover, compared to standard quantization, GIF neurons can be viewed as performing recurrent quantization while preserving SNN neuronal dynamics across time steps.
>
> >Since this approach assigns different values of T corresponding to saliency of particular channels, can you please explain the temporal dynamics of the model (preferably during the on chip 1-bit inference phase) with a schematic diagram? (Q1)
>
> Thank you for your advice.
> We have included both the 1-bit inference and the training dynamics of GIF neurons in Fig. 8 and Fig. 9 in Appendix A.4, respectively.
> - For 1-bit inference, it corresponds to the condition shown in Fig. 8c (please refer to our response to W1).
> - For multi-bit inference, different merged spikes are recurrently encoded using the same input and accumulation equations (Eq. 1) and the output firing equation (Eq. 5). Except for the output firing equation, the processes for 1-bit and multi-bit inferences are identical.

---

> ### Author Response · Authors · 2024-11-22
> **Comparisons with SoTA Quantization**
>
> >Can the authors please compare their results to current sota quantized LLMs. (Q4)
>
> Yes, SpkeLLM focuses on hybrid encoding of both SNNs and quantized-ANNs, and can cooperate with different quantization pipelines including Quantization-Aware Training (QAT), Post-Training Quantization (PTQ), and QLoRA. For comparison, it is crucial to keep similiar training data, and therefore we focus on the most general PTQ pipelines in our main results. We conduct a wide comparison in different pipelines as follows (including BitNet-1.58bit [1] and IR-QLoRA [2]):
>
> (i) Comparison with QLoRA [3] methods:
>
> This line of work focuses on adding a low-rank branch to finetune the low-bit quantized model. State-of-the-art methods including IR-QLoRA [2], QA-LoRA [4], QLoRA [3]. Before training, we calibrate saliency and acquire a 2-bit weight spiking model with the EfficientQAT [5] setting. We follow the same setting of QLoRA and IR-LoRA in Alpaca finetuning. SpikeLLM adding a QLoRA branch exceeds recent methods.
>
> | **Method** | **Data** | **#Bit** | **Hums.** | **STEM** | **Social** | **Other** | **Avg.** |
> |-----------------------------|----------|----------|-----------|----------|------------|-----------|----------|
> | LLaMA-7B | - | 16 | 33.3 | 29.8 | 37.8 | 38.0 | 34.6 |
> | NormalFloat                | -        | 2        | 24.2      | 28.9     | 31.1       | 25.0      | 26.9     |
> | QLoRA (w/ GPTQ)            | Alpaca   | 2        | 23.4      | 26.2     | 26.4       | 28.4      | 25.8     |
> | QLoRA                      | Alpaca   | 2        | 24.0      | 27.0     | 27.5       | 26.7      | 26.2     |
> | QA-LoRA                    | Alpaca   | 2        | 27.3      | 26.1     | 26.1       | 30.3      | 27.5     |
> | IR-QLoRA                   | Alpaca   | 2        | 26.0      | **27.8**     | **30.2**       | 28.3      | 27.8     |
> | **SpikeLLM-QLoRA$_{\texttt{T=2}}$**   | Alpaca   | 2        | **31.5**      | 27.5     | 29.5       | **32.0**      | **30.3**     |
>
> (ii) Comparison with SoTA QAT/PTQ for weight quantization:
>
> For low-bit weight-only quantizations, we train and evaluate SpikeLLM following EfficientQAT [5] (including training data and evaluation metrics, lm-eval-v0.4.2). Both the EfficientQAT and SpikeLLM are trained in 2048 length in two stages with the same condition. We compare the 2-bit performance as follows:
>
> | **Method**              | **#Bit** | **WinoGrande** | **HellaSwag** | **ArcC** | **ArcE** | **PiQA** | **Avg.**  |
> |-------------------------|----------|----------------|---------------|----------|----------|----------|-----------------------|
> | Llama-2-7B                      | -        | 69.22          | 57.16         | 43.52    | 76.26    | 78.07    | 64.85                 |
> | GPTQ                    | 2        | 55.17          | 32.59         | 21.25    | 40.45    | 58.32    | 41.56                 |
> | OmniQ                   | 2        | 55.88          | 40.28         | 23.46    | 50.13    | 65.13    | 46.98                 |
> | AutoRound               | 2        | 61.01          | 40.28         | 32.25    | 65.99    | 72.96    | 54.50                 |
> | AQLM\(_{2x8}\)          | 2        | 65.27          | 49.96         | 32.85    | 66.92    | 73.07    | 57.61                 |
> | EfficientQAT        | 2        | 65.19          | 49.53         | 35.15    | 69.74    | 73.67    | 58.65                 |
> | **SpikeLLM$_{\texttt{T=2}}$** | **2**    | **65.35**      | **50.90**     | **36.01** | **70.54** | **74.54** | **59.47**            |
>
> The comparison of another line of QAT methods are including LLM-QAT [6], BitDistiller [7], EfficientQAT [5]. Because they use various accuracy metrics (arc/arc_norm), we only compare the perplexity (PPL) on Wikitext.
>
> | **Methods**             | **#Bit** | **PPL** |
> |-------------------------|----------|----------------|
> | Llama-2-7B              | FP16     | 5.47           |
> | BitNet-1.58-1.3B        | 1.58     | 11.29          |
> | BitNet-1.58-3.9B        | 1.58     | 9.62           |
> | LLM-QAT                 | 2        | 9.30           |
> | BitDistiller            | 2        | 8.08           |
> | EfficientQAT            | 2        | 7.73           |
> | **SpikeLLM$_{\texttt{T=2}}$**               | 2        | **7.18**           |
>
> (iii) Comparison with SoTA QuaRot for 4W4A quantization:
>
> For weight-activation quantization, one of the most SoTA methods is QuaRot [8] and we compare in **Appendix A.5**. Although QuaRot efficiently overcomes outliers in input layers of transformer modules (key-proj, query-proj, value-proj for self-attention; up-proj, gate-proj for mlp), our crucial finding is that there are still massive outliers (x100000 larger magnitude) in output layers of each module (out-proj for self-attention; down-proj for mlp); moreover, a few attention heads have significant saliency and magnitude which QuaRot can not direct address by Hadamard transformations (because of different softmax attention weights).
>
> ----------- to continue -------------

---

> ### Author Response · Authors · 2024-11-22
> **Comparisons with SoTA Quantization**
>
> Based on the above observations, we apply saliency-aware spiking steps and OBSpiking only in out-proj for self-attention and down-proj for mlp. When training and evaluating with QuaRot pipeline, SpikeLLM could improve QuaRot performance and better address the outliers in LLMs (**Appendix A.5**).
>
> | **Method**         | **Saliency** | **#Bits** | **PIQA$_{\text{norm}}$** | **ARC-e$_{\text{norm}}$** | **ARC-c$_{\text{norm}}$** | **BoolQ** | **HellaSwag$_{\text{norm}}$** | **Winogrande** | **Avg.**  |
> |--------------------|--------------|-----------|--------------------------|---------------------------|---------------------------|-----------|------------------------------|----------------|-----------|
> | Llama-2-7b         | -            | FP16      | 79.11                    | 74.54                     | 46.42                     | 77.77     | 75.99                        | 69.06          | 70.48     |
> | QuaRot-RTN         | -            | W4A4      | 71.82                    | 59.89                     | 36.18                     | 67.37     | 63.88                        | 59.12          | 59.71     |
> | SpikeLLM-RTN       | 0.07         | W4A4      | 73.07                    | 61.99                     | 36.26                     | 68.81     | 64.15                        | 59.12          | **60.57** |
> | QuaRot-GPTQ        | -            | W4A4      | 75.95                    | 68.43                     | 39.76                     | 72.54     | 72.23                        | 64.72          | 65.61     |
> | SpikeLLM-GPTQ      | 0.08         | W4A4      | 76.77                    | 69.91                     | 42.06                     | 72.39     | 72.30                        | 65.43          | **66.48** |
> | Llama-2-13B        | -            | FP16      | 80.63                    | 77.48                     | 49.23                     | 80.73     | 79.37                        | 71.74          | 80.69     |
> | QuaRot-RTN         | -            | W4A4      | 74.86                    | 69.19                     | 41.98                     | 72.54     | 70.35                        | 64.72          | 65.61     |
> | SpikeLLM-RTN       | 0.05         | W4A4      | 75.35                    | 69.19                     | 43.00                     | 73.09     | 70.73                        | 66.46          | **66.30** |
> | QuaRot-GPTQ        | -            | W4A4      | 77.91                    | 72.18                     | 46.16                     | 78.41     | 75.55                        | 68.82          | 69.84     |
> | SpikeLLM-GPTQ      | 0.04         | W4A4      | 78.51                    | 71.89                     | 47.27                     | 79.02     | 75.77                        | 69.38          | **70.31** |
>
> Based on these results, the performance of SpikeLLM is able to match state-of-the-art quantization methods. Moreover, it also addresses the generalization ability under different training data and pipelines.
>
> >The temporal dynamics of the model is not explained in details. There is very limited results on the activity sparsity of the model, energy-efficiency comparison with other efficient LLMs (Table 5 only compares with LLAMA-1-7B) (W1)
>
> Thank you for the advice. We additionally calculate the speed-up of Llama-1-7B, Llama-2-7B, Llama2-13B in 4W4A settings in 1-bit inference; we calculate speed-up according to ACE metrix [8].
>
> | Model         | Llama-1-7B | Llama-2-7B | Llama-2-13B | Llama-1-7B | Llama-2-7B | Llama-2-13B |
> |---------------|------------|------------|-------------|------------|------------|-------------|
> | Saliency Step | 2          | 2          | 2           | 4          | 4          | 4           |
> | FP16          | 3886.22    | 3886.22    | 7507.60     | 3886.22    | 3886.22    | 7507.60     |
> | Binary Infer. | 630.83     | 608.78     | 1167.48     | 648.09     | 625.83     | 1204.36     |
> | Speed-up      | x6.16      | x6.38      | x6.43       | x6.00      | x6.21      | x6.23       |
>
> >In Table 6, SpikeLLM uses ternary GIF neurons as weight quantizers. What is the quantization level of the activations? (Q2)
>
> We follow the PB-LLM [8] settings in Table 6 and don't quantize activations. If one of the weights or activations are binary or ternary, the linear layer could implemented as additions.
>
> >The authors can use better baselines for comparison such as 1.58 bit LLM [1], IR-QLoRA[2]. (W2)
>
> The comparisons with IR-LoRA could refer to the response of Q4. Since the official version of BitNet-1.58bit is not open-sourced and different testing methods use distinct zero-shot prompts, our comparison in Q4 focuses solely on the Wikitext-PPL results. (If necessary, we can further compare using the data reported in the paper.)
>
> ----------- to continue -------------

---

> ### Author Response · Authors · 2024-11-22
> **Comparisons with SoTA Quantization**
>
> In training methods, BitNet-1.58bit is trained from scratch, which limits the pretraining model size under constrained computational resources. The largest BitNet-1.56bit model is 3.9B.
>
> SpikeLLM aims to achieve two key objectives:
> - Leverage spiking neuron dynamics to enhance current quantization performance, making it broadly applicable to PTQ, QAT, and QLoRA scenarios.
> - Propose a spiking large model capable of inference in a 1-bit working mode, scaling SNNs to over 10B parameters.
>
> **Reference**
>
> [1] Ma S, Wang H, Ma L, et al. The era of 1-bit llms: All large language models are in 1.58 bits[J]. arXiv preprint arXiv:2402.17764, 2024.
>
> [2] Qin H, Ma X, Zheng X, et al. Accurate lora-finetuning quantization of llms via information retention[J]. arXiv preprint arXiv:2402.05445, 2024.
>
> [3] Dettmers T, Pagnoni A, Holtzman A, et al. Qlora: Efficient finetuning of quantized llms[J]. Advances in Neural Information Processing Systems, 2024, 36.
>
> [4] Xu Y, Xie L, Gu X, et al. Qa-lora: Quantization-aware low-rank adaptation of large language models[J]. arXiv preprint arXiv:2309.14717, 2023.
>
> [5] Chen M, Shao W, Xu P, et al. Efficientqat: Efficient quantization-aware training for large language models[J]. arXiv preprint arXiv:2407.11062, 2024.
>
> [6] Liu Z, Oguz B, Zhao C, et al. Llm-qat: Data-free quantization aware training for large language models[J]. arXiv preprint arXiv:2305.17888, 2023.
>
> [7] Du D, Zhang Y, Cao S, et al. Bitdistiller: Unleashing the potential of sub-4-bit llms via self-distillation[J]. arXiv preprint arXiv:2402.10631, 2024.
>
> [8] Ashkboos S, Mohtashami A, Croci M L, et al. Quarot: Outlier-free 4-bit inference in rotated llms[J]. arXiv preprint arXiv:2404.00456, 2024.

---

> > ### Comment · Reviewer_wFix · 2024-11-24
> >
> > Thank you for the detailed response. It addresses majority of my concerns. I am raising my score to 6.

---

### Meta-Review · Area_Chair_XLcr · 2024-12-19

**Metareview:**

The work claims to propose the first spiking LLM, and considers models with up to 70B parameters. Improvements in perplexity, accuracy in common-sense reasoning when compared to quantized LLMs are claimed. Moreover, up to 10-fold reductions in computation are achieved.

The paper addresses an important problem and the experimental evaluation follows the standards of the LLM community, and the claims are validated with comparisons to recently published baselines.

The paper makes an important contribution in new methods to improve energy efficiency of LLMs, and will be of interest to the ICLR community. I therefore recommend acceptance.

**Additional Comments On Reviewer Discussion:**

All reviewers appreciated the contributions and found the main claims of the paper to be validated in thorough numerical experiments against strong baselines. Reviewer 2u4J had concerns about the computational complexity, which were addressed in the rebuttal phase.

---

### Decision · Program_Chairs · 2025-01-22

Accept (Poster)